# CODA: COORDINATING THE CEREBRUM AND CERE-BELLUM FOR A DUAL-BRAIN COMPUTER USE AGENT WITH DECOUPLED REINFORCEMENT LEARNING.

## ABSTRACT

Autonomous agents for Graphical User Interfaces (GUIs) face significant challenges in novel software, require both long-horizon planning with software domain knowledge and precise, fine-grained execution. Existing approaches suffer from a trade-off: generalist agents excel at planning but falter in execution, while specialized agents show the opposite weakness. While recent compositional frameworks attempt to bridge this gap by combining a "planner" and an "actor", they are typically static and non-trainable, preventing adaptation from experience—a critical limitation given the scarcity of high-quality data in novel software. To address these limitations, we introduce **CODA**, a novel and trainable compositional framework that synergizes a generalist planner (Cerebrum) with a specialist executor (Cerebellum), trained with a dedicated two-stage training pipeline. The first stage, **Specialization**, employs a decoupled GRPO approach to train an expert planner for each novel software individually. The second stage, **Generalization**, aggregates all positive trajectories from all specialized experts. This consolidated, high-quality dataset is then used to perform supervised fine-tuning (SFT) on the final planner, equipping it with the robust, cross-domain capabilities of a generalist. Evaluated ScienceBoard benchmark with diversified novel softwares, our framework significantly outperforms the baseline and establishes a new state-of-the-art (SOTA) among open-source models with strong generalizability to novel software and unseen executor like code agent. All the code and models will be made publicly available to foster further research.

## 1 INTRODUCTION

Autonomous agents for Graphical User Interfaces (GUIs) (Anthropic, 2024; OpenAI, 2025; Qin et al., 2025; Lin et al., 2024; Wu et al., 2024b; Hong et al., 2023) promise to automate a wide range of digital tasks (Zhou et al., 2023; Xie et al., 2024). However, their application in specialized domains such as scientific computing and engineering analysis remains highly challenging (Sun et al., 2025a). These environments pose two primary difficulties: first, their interfaces are highly complex, requiring precise and fine-grained actions; second, the problems they address are intrinsically complicated, demanding long-horizon planning to achieve effective solutions.

Effective agency for computer task automation in these domains requires both high-level planning and low-level execution as well as domain knowledge. However, current models exhibit a clear trade-off. Generalist models like Qwen2.5-VL (Bai et al., 2025) provide robust planning capabilities but often struggle with the precise grounding needed for reliable execution. Conversely, specialized agents (Wu et al., 2024b; 2025; Xie et al., 2025) like UI-Tars (Qin et al., 2025) are highly proficient in execution, yet lack of domain specific knowledge for novel knowledge and require human labeled trajectories on these softwares for fine-tuning.

To bridge this gap, a natural approach has been to develop compositional frameworks that explicitly decouple planning from execution, effectively pairing a generalist "cerebrum" with a specialist "cerebellum" (Agashe et al., 2024; 2025; Song et al., 2025). While promising, these pioneering approaches are fundamentally limited. They are typically static and non-trainable, relying on powerful, often closed-source models as their core planner. This design introduces significant drawbacks: it

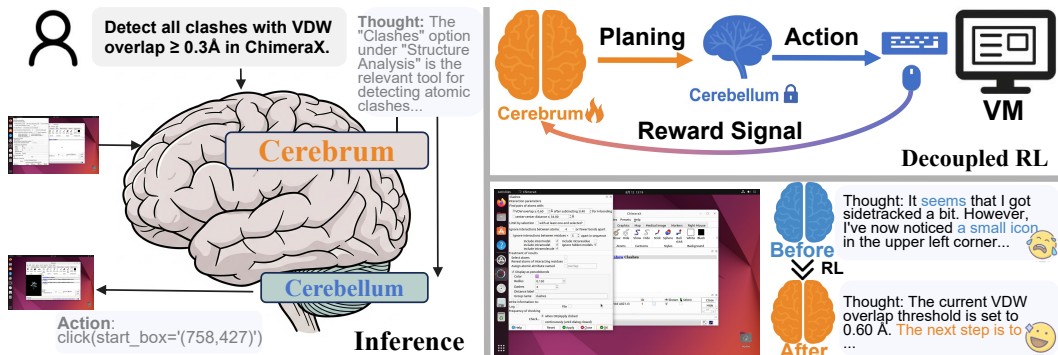

Figure 1: **Overall architecture of the proposed learnable Planner–Executor framework.** Analogous to the relationship between the cerebrum and the cerebellum in the human brain, the Planner (cerebrum) generates high-level thoughts based on the history and screenshots, while the Executor (cerebellum) executes concrete GUI actions accordingly.

compromises transparency and replicability, and most critically, prevents the agent from learning and adapting through experience on novel softwares.

This architectural decoupling is not merely an engineering convenience but is inspired by the functional architecture of the human brain (illustrated in Fig.1). The specialization of high-level planning (the Cerebrum) and low-level motor control (the Cerebellum) is a key aspect of human intelligence. Crucially, these structures exhibit different learning patterns: the Cerebellum, once mature, provides stable and broadly applicable motor skills that require infrequent updates (Ito, 2000). In contrast, the Cerebrum continuously adapts its strategies based on the nuances of new tasks and environments (Demarin & Morović, 2014; Hallett, 2005). This biological parallel motivates our core hypothesis: an effective agent should pair a stable, proficient grounding model with a dynamic planner that is specialized for different software domains through targeted, experience-driven learning.

To realize this vision, we propose a trainable compositional framework that integrates Qwen2.5-VL (Bai et al., 2025) as the planner (cerebrum) and UI-Tars-1.5 (Qin et al., 2025) or GUI-Actor (Wu et al., 2025) as the executor (cerebellum). Unlike prompting-based systems that rely on proprietary closed-source planners, our framework makes the planner itself learnable through interaction with software environments mediated by a static executor. Concretely, the executor provides stable, software-agnostic grounding for low-level GUI actions, while the planner, by leveraging this reliable interface, can gradually acquire domain-specific knowledge and improve its high-level planning strategies. In contrast to end-to-end training of a single agent, which requires massive amounts of specialized data and costly retraining of both perception and execution modules, our decoupled approach is substantially more data efficient: only the planner is optimized for domain adaptation, while the executor remains fixed as a general-purpose grounder that already possesses strong generalization ability after massive pretraining for grounding purposes. This design reduces reliance on curated trajectories, lowers training cost, and ensures controllable adaptation.

To train the planner effectively under this cerebrum–cerebellum separation, we avoid the need for costly human-labeled trajectories. Instead, we leverage a judging system built from open-source models to automatically provide dense reward signals, combined with autonomous interaction with scientific software environments through the static executor. This setup enables the planner to gradually acquire domain-specific planning ability with zero human effort. Furthermore, by distributing the interaction process across multiple software environments in parallel—coordinated by a central master—we can significantly accelerate reinforcement learning. This strategy not only makes the training process more efficient but also echoes our brain-inspired design: the cerebellum-like executor delivers stable grounding, while the cerebrum-like planner continually adapts through experience. Our work makes three major contributions:

1. We propose a decoupled reinforcement learning framework that pairs a learnable planner with a fixed executor, enabling efficient domain adaptation by isolating strategic learning from low-level grounding and control.

2. We propose a fully automated judging system using a fine-tuned model and voting strategy to generate dense reward signals, which removes the dependency on human-annotated data for

training. This judging system achieve strong performance on AgentRewardBench (Lù et al., 2025) and other software environments.

3. Planner trained via our framework based on Qwen2.5-VL (Bai et al., 2025) achieve strong performance on the ScienceBoard (Sun et al., 2025a) benchmark with generalizable adaption on OSWorld (Xie et al., 2024) and novel executor like code agent.

## 2 RELATED WORKS

**Reinforcement Learning for LVLMs (Large Vision Language Models).** Training for LLMs(Large Language Models) and LVLMs (Touvron et al., 2023; Grattafiori et al., 2024; Liu et al., 2023a; Bai et al., 2025; Wang et al., 2024; Xing et al., 2025; Sun et al., 2024d;c; Ding et al., 2025) has progressed from data-intensive Supervised Fine-Tuning (SFT) (Liu et al., 2023a; Wei et al., 2022) towards Reinforcement Learning (RL). Algorithms like Group Relative Policy Optimization (GRPO) (Guo et al., 2025; Shao et al., 2024) have proven effective for reasoning tasks, moving beyond earlier single-turn RLHF applications (Ouyang et al., 2022; Ziegler et al., 2019; Rafailov et al., 2023). However, applying RL to complex agentic tasks (Bai et al., 2024; Qi et al., 2024; Zhou et al., 2024; Zhai et al., 2024; Carta et al., 2023) is challenging. Prevailing methods train monolithic agents end-to-end, often requiring co-trained critic models (Schulman et al., 2015) or preference-based optimization like DPO (Rafailov et al., 2023; Putta et al., 2024; Qin et al., 2025), which problematically entangles the distinct skills of planning and execution. In contrast, our work employs a decoupled reinforcement learning strategy: the high-level planner is optimized via environmental interaction while the execution model remains fixed. We adapt GRPO by computing rewards from the reward system and backpropagating the advantage exclusively through planning tokens. This targeted optimization stably enhances strategic planning, distinguishing our method from prior works that train dedicated critic models (Bai et al., 2024; Qi et al., 2024; Wang et al., 2025) or use filtered behavior cloning (Pan et al., 2024; Chen et al., 2020).

**Computer Use Agent.** Fueled by advancements in Large Vision-Language Models (LVLMs) (Touvron et al., 2023; Grattafiori et al., 2024; Liu et al., 2023a; Bai et al., 2025; Wang et al., 2024), a new generation of agents capable of operating computers via multi-modal inputs is emerging (Hu et al., 2024b; Hong et al., 2024; Cheng et al., 2024; Nguyen et al., 2024; Lin et al., 2024; Sun et al., 2024b). Whether processing structured text and code (Qi et al., 2024; Putta et al., 2024; Lai et al., 2024; Sun et al., 2024a; Nakano et al., 2021) or screenshots (Hong et al., 2023; Lin et al., 2024; Wu et al., 2024b; OpenAI, 2025), these agents face an inherent dichotomy analogous to human cognition: the tension between high-level strategic planning and precise, low-level action execution. This has motivated the development of compositional frameworks that decouple these responsibilities (Agashe et al., 2024; 2025; Liu et al., 2023b; Zhang et al., 2025; Song et al., 2025). However, a significant portion of this research relies on static, non-trainable systems that orchestrate powerful, often proprietary models (Anthropic, 2024; OpenAI, 2025; Google DeepMind, 2025; Yan et al., 2023; He et al., 2024; Zhang et al., 2024; Wang et al., 2023; Wu et al., 2024a) as their core planner. This design fundamentally prevents the agent from adapting through experience—a critical flaw for mastering novel software where interaction data is scarce. Our work charts a different course by exploring reinforcement fine-tuning of the planner. By enabling the planner to learn specialized domain knowledge through direct software interaction via a fixed execution model, our strategy achieves robust performance on unfamiliar applications.

## 3 METHOD

### 3.1 PROBLEM FORMULATION

We formally define the task of autonomous GUI operation for software workflows as a Partially Observable Markov Decision Process (POMDP). Each task is initiated with a natural language instruction $g$ from the task space $\mathcal{G}$. At each timestep $t$, the agent perceives the latent environment state $s_t \in \mathcal{S}$ through a visual observation $o_t \in \Omega$, consisting of a screenshot of the user interface. The agent's behavior is governed by a policy $\pi$, instantiated by a large vision-language model, which synthesizes an action program $a_t \in \mathcal{A}$. The action space $\mathcal{A}$ consists of precisely parameterized `pyautogui` scripts, where precision in arguments (e.g., coordinates) is critical for execution. The

Figure 2: **Overall training process of the proposed Planner–Executor framework.** The Planner generates high-level thoughts based on the history and screenshots, while the Executor executes concrete GUI actions accordingly. During training, the rewards are calculated from $a^{(i)}$ and applied to $p^{(i)}$ to calculate loss.

policy generates this action based on the initial instruction and the history of interactions:

$$a_t = \pi(g, (o_1, a_1, \ldots, a_{t-1}, o_t))$$

This sequential process induces a state trajectory $\tau = (s_0, s_1, \ldots, s_T)$ with the maximum time step $T$. A task is considered successful if the final state $s_T$ satisfies the predefined goal condition specified in $\mathcal{G}$.

## 3.2 MODEL ARCHITECTURE

To address the inherent trade-off in monolithic models, which struggle to balance long-horizon planning with precise action grounding, we propose a composite agent architecture that structures the decision-making process into a Planner-Executor framework. This design decouples the task into two distinct yet collaborative modules: a high-level Planner responsible for strategic thinking and a low-level Executor for concrete action execution.

**Planner** The Planner is instantiated from the Qwen2.5-VL (Bai et al., 2025) model. Its primary responsibility is to analyze the task's progress and formulate a high-level, explicit plan $p_t$ for each step. Specifically, at each timestep $t$, the Planner receives the interaction history up to the previous step $m_{t-1} = (p_1, a_1, \ldots, p_{t-1}, a_{t-1})$, the current visual observation $o_t$, and the preceding observation $o_{t-1}$. The output is a structured thought, denoted as $p_t$, which outlines the immediate objective and explicitly identifies the target UI elements for interaction. The process can be summarized as:

$$p_t = \text{Planner}(m_{t-1}, o_{t-1}, o_t)$$

**Executor** The Executor employs a specialized GUI groudning model (UI-TARS-1.5 (Qin et al., 2025) or GUI-Actor (Wu et al., 2025)). Its role is to translate the Planner's abstract thought $p_t$ into a precise, executable action. The Executor is provided with the same historical and visual context as the Planner ($m_{t-1}$, $o_{t-1}$, and $o_t$), but is critically augmented with the Planner's newly generated thought $p_t$. Its output is a low-level GUI action $a_t$ in the form of a 'pyautogui' command, such as 'click(x, y)'. The Executor's operation is defined as:

$$a_t = \text{Executor}(m_{t-1}, o_{t-1}, o_t, p_t)$$

## 3.3 TRAINING PIPELINE

Our training methodology employs a two-stage curriculum designed for initial specialization followed by broad generalization.

### 3.3.1 STAGE 1: SPECIALIZATION VIA DECOUPLED REINFORCEMENT LEARNING

The primary objective of this initial training stage is to enhance the agent's specialized performance on individual software applications.

Through empirical analysis, we observed that the Executor exhibits strong generalization capabilities, accurately translating well-structured plans into executable actions. However, the Planner

module emerged as the primary bottleneck, often struggling to formulate effective high-level strategies. To address this, we adopt a decoupled training strategy that focuses reinforcement learning exclusively on the Planner ($\pi_\theta$ = Planner). This targeted approach allows us to refine the agent's strategic reasoning without altering the already competent Executor.

Since the initial Planner is relatively weak and generates a limited number of successful trajectories, standard reinforcement learning methods can be inefficient. Therefore, we adapt the Group Relative Policy Optimization (GRPO) framework (Guo et al., 2025; Shao et al., 2024), which is particularly effective in such scenarios. GRPO can derive a meaningful learning signal by comparing the relative quality of different outputs, even when most of them are suboptimal.

The training process for a given task unfolds as follows. Given the current state and interaction history, the Planner first generates a group of $G$ candidate plans. Subsequently, the fixed Executor takes each plan as input and produces a corresponding low-level action. To generate a fine-grained learning signal, we compute a reward for each plan by comparing its resulting action $a^{(i)}$ to the labeled positive action $a_T$ (details of labeling process are in Sec.3.4 ). Our composite reward function assesses both the correctness of the action type and the precision of its parameters:

$$r^{(i)} = r(a^{(i)}, a_T) = \mathbb{I}\left(\text{type}(a^{(i)}) = \text{type}(a_T)\right) + r_{\text{dist}}(a^{(i)}, a_T), \tag{1}$$

Here, the indicator function $\mathbb{I}(\cdot)$ provides a binary reward for selecting the correct type of action (e.g., `click` vs. `type`). The term $r_{\text{dist}}(a^{(i)}, a_T)$ offers a continuous reward based on the parametric similarity between the predicted and ground-truth actions, such as L1 distance for coordinates or IoU for bounding boxes. These distance-based rewards are normalized to $[0, 1]$ to ensure consistent scaling.

Once the rewards are calculated, they are used to derive a relative advantage $A^{(i)}$ for each plan, which is then fed into the GRPO loss function to update the Planner policy:

$$A^{(i)} = \frac{r^{(i)} - \text{mean}(\{r^{(j)}\}_{j=1}^G)}{\text{std}(\{r^{(j)}\}_{j=1}^G)}, \quad i = 1, \cdots, G. \tag{2}$$

The GRPO loss is formulated as follows:

$$\mathcal{L}_{\text{GRPO}}(\pi_\theta) = -\mathbb{E}_{(s,I)\sim\mathcal{D}, \{a^{(i)}\}_{i=1}^G \sim \pi_{\text{ref}}(\cdot|s,I)} \tag{3}$$

$$\left[\frac{1}{G}\sum_{i=1}^G \frac{1}{|p^{(i)}|} \sum_{t=1}^{|p^{(i)}|} \left\{ \min\left(r_t^{(i)}(\theta)A^{(i)}, \text{clip}(r_t^{(i)}(\theta), 1-\epsilon, 1+\epsilon)A^{(i)}\right) - \beta\, D_{\text{KL}}^{(i,t)}(\pi_\theta\|\pi_{\text{ref}})\right\}\right],$$

where $\quad r^{i,t}(\theta) = \frac{\pi_\theta(p^{(i)}|s,I)}{\pi_{\theta_{\text{ref}}}(p^{(i)}|s,I)}$ and $\quad D_{\text{KL}}^{i,t}(\pi_\theta, \pi_{\text{ref}}) = \frac{\pi_{\text{ref}}(p^{(i)}|s,I)}{\pi_\theta(p^{(i)}|s,I)} - 1 - \log\frac{\pi_{\text{ref}}(p^{(i)}|s,I)}{\pi_\theta(p^{(i)}|s,I)}.$

Consistent with the approach in (Shao et al., 2024; Guo et al., 2025), this advantage is applied across all reasoning tokens in the plan $p^{(i)}$, encouraging the model to develop more robust and free-form planning capabilities.

### 3.3.2 STAGE 2: GENERALIZATION VIA AGGREGATED SUPERVISED FINE-TUNING

We adopt the specialist-to-generalist paradigm proposed in (Sun et al., 2025b), where a generalist model is trained by leveraging multiple specialist models as teachers. We observe that directly applying reinforcement learning across all software leads to suboptimal performance. To address this, we first train four specialist models using the methods described in Sec. 3.3.1. These specialists are then employed to generate new trajectories for each software, which serve as supervision for training a generalist model. After learning from the four software-specific teachers, the resulting generalist not only surpasses its teachers in performance, but also demonstrates stronger reasoning and reflection abilities during planning, as well as broader domain knowledge across different software.

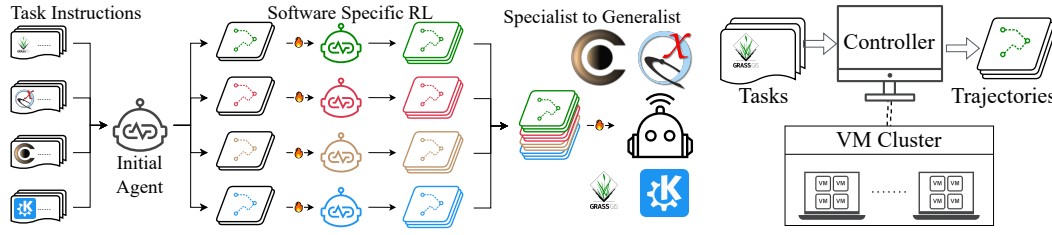

(a) Specialist-to-Generalist strategy.      (b) Distributed VM System.

Figure 3: Specialist-to-generalist as show in (a) and distributed virtual machine system in (b).

## 3.4 AUTO EXPLORATION PIPELINE.

**Auto Task Generation.** To generate tasks, we employ Qwen2.5-72B (Wang et al., 2024) to produce a diverse set of high-level objectives, with the detailed prompt provided in Fig. 12. Our training methodology proceeds in two stages. The first stage is decoupled reinforcement learning, where the agent repeatedly interacts with software environments to execute the generated tasks. During this process, our judge system provides the necessary reward signals to guide the agent's learning. In the second stage, aggregated fine-tuning, we leverage the specialized agent from the first stage to collect a diverse dataset of interaction trajectories. These trajectories are filtered by the judge system to retain only those containing positive outcomes, which are then used for supervised fine-tuning.

**Judge System for Providing Reward Signals.** Our judge system labels the positive actions $a_T$ within an agent's trajectory when performing a task. Given a full trajectory $\mathcal{H} = \{o_0, a_0, \ldots, o_{\text{final}}\}$, the judge takes the complete sequence of screenshot observations $(o_1, o_2, \ldots, o_n)$ as input and outputs three signals: `Correctness`, `Redundant`, and `First Error Step`, using the detailed prompt shown in Fig. 13. A trajectory is considered clean and successful when `Correctness` is `True` and both `Redundant` and `First Error Step` are empty. In this case, all actions $a$ in the trajectory are labeled as $a_T$. We present a detailed evaluation of the judge's precision and discuss approaches for improving it via multi-resolution voting and model ensembling in Sec. 4.2.

**Distributed Virtual Machine System.** Task execution is the most time-consuming step in our pipeline, so we developed a lightweight distributed system to accelerate large-scale parallel interaction and trajectory curation. As illustrated in Fig. 3b, the system follows an HTTP-based master–client architecture: the master node manages a dynamic task queue, monitors execution progress, and aggregates results, while multiple client nodes execute tasks in parallel within isolated virtual machine environments. This design enables efficient scaling to hundreds of concurrent environments, substantially reducing the time required to collect successful trajectories and making the framework well-suited for large-scale training and evaluation.

## 4 EXPERIMENTS

### 4.1 AGENT PERFORMANCE EVALUATION.

Our planner-executor approach is based on Qwen2.5VL-32B (Bai et al., 2025) serve as planner and UI-TARS-1.5-7B (Qin et al., 2025) or GUI-Actor (Wu et al., 2025) serve as executor. We use method proposed in Sec.3.4 to generate high level tasks for each software from ScienceBoard (Sun et al., 2025a) through decoupled reinforcement learning proposed in Sec.3.3.1. During Training, the reward signal is provided by our judge system evaluated in Sec.4.2. Our Training is done for 500 iterations on 32 NVIDIA A100 80G GPUs, with a batch size of 16 and a learning rate of $2 \times 10^{-5}$ scheduled via cosine decay based on OpenRLHF (Hu et al., 2024a). Instruction statistics is detailed in Sup. C. As reported in Tab.1 noted as CODA (Stage-1), our evaluation is done on four GUI centric software from ScienceBoard (Sun et al., 2025a). We also report other planner-executor decoupled approaches. This first-stage reinforcement learning approach lead to significant performance gain compared to baseline.

In the second stage, we consolidate the expertise from the specialized planners into a single, powerful generalist model using the specialist-to-generalist fine-tuning method proposed in Sec. 3.3.2. The individual specialist planners serve as "teacher" models to generate a dataset of 5.6K multi-turn interaction trajectories, which are subsequently filtered by our judge system to ensure data quality

Table 1: Success rates of various models on ScienceBoard (Sun et al., 2025a). Proprietary models and open-sourced models based methods are highlighted with purple and green backgrounds, respectively. *Indicates specialist agents trained separately for each software with ensembled results.

| Metrics | Model | Success Rate (↑) | | | | |
|---------|-------|---------|---------|-----|--------|---------|
| | | **Algebra** | **Biochem** | **GIS** | **Astron** | **Overall** |
| Average@1 | GPT-4o (OpenAI, 2023) | 3.23% | 0.00% | 0.00% | 0.00% | 0.81% |
| | Claude-3.7-Sonnet (Anthropic, 2025) | 9.67% | 37.93% | 2.94% | 6.06% | 14.15% |
| | Gemini-2.0-Flash (Team et al., 2023) | 6.45% | 3.45% | 2.94% | 6.06% | 4.73% |
| | GPT4o→UGround-V1-7B (Gou et al., 2024) | 0.00% | 3.45% | 0.00% | 3.03% | 1.62% |
| | GPT4o→OS-Atlas-Pro-7B (Wu et al., 2024b) | 6.25% | 10.34% | 0.0% | 3.03% | 4.92% |
| | GPT4o→UI-TARS-72B (Qin et al., 2025) | 3.23% | 10.34% | 5.88% | 6.06% | 6.38% |
| | Qwen2.5-VL-72B (Bai et al., 2025) | 22.58% | 27.59% | 5.88% | 9.09% | 12.94% |
| | InternVL3-78B (Zhu et al., 2025) | 6.45% | 3.45% | 0.00% | 0.00% | 2.69% |
| | UI-TARS-1.5-7B (Qin et al., 2025) | 12.90% | 13.79% | 0.00% | 6.06% | 8.19% |
| Average@8 | Qwen2.5-VL-32B (Bai et al., 2025) | 10.48% | 13.79% | 1.47% | 4.55% | 7.57% |
| | UI-TARS-1.5-7B (Qin et al., 2025) | 6.49% | 10.24% | 0.80% | 3.03% | 5.14% |
| | **CODA +Actor(Stage-1)*** | 12.90% | 30.17% | 10.66% | 7.58% | 15.33% |
| | **CODA +Actor(Stage-2)** | 18.55% | 31.03% | **16.54%** | 11.74% | 19.47% |
| | **CODA +TARS-1.5(Stage-1)*** | 13.71% | 26.29% | 7.72% | 9.85% | 14.39% |
| | **CODA +TARS-1.5(Stage-2)** | **20.16%** | **32.23%** | 14.71% | **17.05%** | **21.04%** |
| Pass@8 | Qwen2.5-VL-32B (Bai et al., 2025) | 29.03% | 31.03% | 8.82% | 9.09% | 19.49% |
| | UI-TARS-1.5-7B (Qin et al., 2025) | 19.35% | 24.14% | 5.88% | 12.12% | 15.36% |
| | **CODA +Actor(Stage-1)*** | 32.25% | 34.48% | 26.47% | 21.21% | 28.61% |
| | **CODA +Actor(Stage-2)** | 38.71% | 37.93% | **32.35%** | 24.24% | 33.31% |
| | **CODA +TARS-1.5(Stage-1)*** | 41.94% | 44.83% | 23.53% | 18.18% | 32.12% |
| | **CODA +TARS-1.5(Stage-2)** | **48.39%** | **51.72%** | 29.41% | **30.30%** | **39.96%** |

Table 2: Evaluation of different judge methods on AgentRewardBench (Lù et al., 2025), OS-World (Xie et al., 2024), and ScienceBoard (Sun et al., 2025a).

| Method | AgentRewardBench (Lù et al., 2025) | | OSWorld (Xie et al., 2024) | | ScienceBoard (Xie et al., 2024) | |
|--------|-----------|--------|-----------|--------|-----------|--------|
| | Precision | Recall | Precision | Recall | Precision | Recall |
| Qwen2.5-VL-72B | 64.5 | 83.4 | 41.5 | 76.9 | 30.2 | 80.1 |
| 72B-GUI-Judge | 73.5 | 79.0 | 76.2 | 79.4 | 63.7 | 80.1 |
| 72B-voting@4 | 76.1 | 79.5 | 63.1 | 73.2 | 58.6 | 75.3 |
| 72B-voting@4 w/ multi-res | 78.9 | 77.4 | 70.3 | 75.3 | 65.7 | 77.9 |
| 72B-voting@4 Ensemble | 81.2 | 76.8 | 79.1 | 73.0 | 69.5 | 74.2 |

(details in Sup. C). A new generalist planner, also initialized from Qwen2.5VL-32B, is then trained on this aggregated dataset. The supervised fine-tuning process ran for 500 iterations on 32 NVIDIA A100 80G GPUs, with a batch size of 64 and a learning rate of $1 \times 10^{-5}$. As shown in Table 1, our resulting model, denoted CODA (Stage-2), surpasses the performance of an ensemble of the individual specialist agents, demonstrating significantly enhanced reasoning and planning capabilities. This result validates the effectiveness of our specialist-to-generalist knowledge transfer strategy. In contrast, a direct reinforcement learning approach across all four software environments with an equivalent training budget led to a compromised success rate, performing worse than the ensemble of the four specialist agents. This comparison strongly underscores the superiority of our two-stage specialist to generalist training paradigm over a monolithic training approach.

## 4.2 TOWARDS PRECISE JUDGING SYSTEM

Our reinforcement learning framework heavily relies on accurate judgments of agent trajectories to provide reliable reward signals. In this section, we present a detailed evaluation of our judge model, which demonstrates improved precision in decision making.

**Settings.** We conduct experiments on two sources of trajectories. (1) AgentRewardBench (Lù et al., 2025), a benchmark designed specifically for judge evaluation. (2) A trajectory dataset we collected from ScienceBoard (Sun et al., 2025a). We run Qwen2.5-VL-72B (Bai et al., 2025) on ScienceBoard tasks and extract 377 labeled trajectories, which are then used as inputs to our judge model. This setup allows us to quantitatively assess the judge's ability to discriminate between successful and failed executions. We report *Precision* and *Recall* as our primary metrics. For voting-based strategies, we adopt a sampling temperature of $T = 1.0$ and a nucleus sampling probability of $top\_p = 0.6$ over 4 independent inference runs.

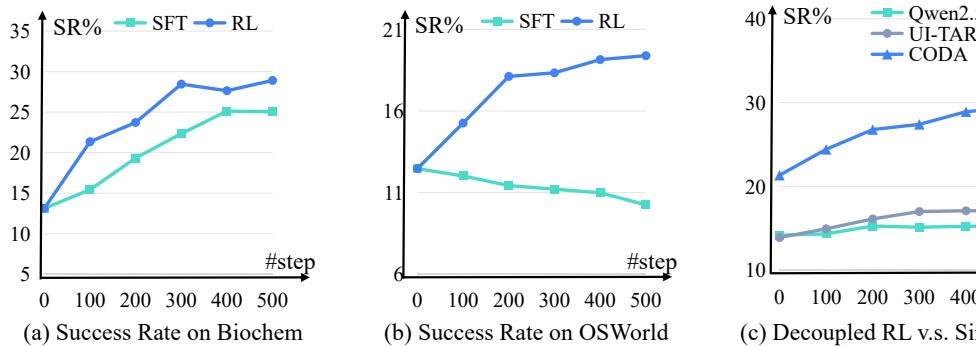

Figure 4: Success Rate During training comparing RL with SFT (a, b) and single model RL (c).

**Results.** As summarized in Tab. 2, our evaluations reveal three effective strategies for improving precision, building upon difference description fine-tuning (Sun et al., 2025b): 1. *Voting.* Instead of a single query, we prompt the model multiple times with high randomness ($T = 1.0$, $top\_p = 0.6$). A trajectory is only deemed successful if all votes agree, which significantly reduces false positives. 2. *Multi-resolution inputs.* Trajectories often include long sequences of high-resolution screenshots. We observe that using a mixture of resolutions across voting rounds is beneficial: low-resolution images help capture global execution dynamics, while high-resolution images aid in detecting fine-grained correctness. In practice, we first apply low-resolution inputs to quickly filter out failures, thereby improving both precision and efficiency. 3. *Model ensembling.* In addition to the fine-tuned judge model (detailed in Sup. B), we find that ensembling two models within the voting strategy further enhances precision.

Across both ScienceBoard (Sun et al., 2025a) and AgentRewardBench (Lù et al., 2025), we observe a consistent progression: the fine-tuned model (72B-GUI-Judge) primarily improves judging precision; multi-resolution inputs add further gains, and ensembling achieves the best balance with the highest precision while maintaining competitive recall. This consistent trend across benchmarks highlights the robustness and generality of our proposed strategies. With methods proposed in Sec. 3.3.1. This judge system provide high quality reward signal for the planner to perform RL to improve reasoning ability and learning software domain knowledge via real interaction.

### 4.3 ABLATION STUDIES.

We conduct a comprehensive ablation study, presented in Tab. 3, to validate the contribution of each component and evaluate the generalizability of our method. For this study, models are trained on the BioChem software from ScienceBoard (Sun et al., 2025a) and tested on the OSWorld (Xie et al., 2024). Our analysis of different reward signals reveals that directly employing a powerful base model (Qwen2.5-VL-72B (Bai et al., 2025)) as the re-

Table 3: **Success Rate** on in-domain BioChem from ScienceBoard (Sun et al., 2025a) and out-of-domain OSWorld (Xie et al., 2024).

| Model | | BioChem | OSWorld |
|---|---|---|---|
| Baseline | Qwen2.5VL+GUI-Actor | 17.24 | 12.47 |
| Reward From | Qwen2.5-VL-72B | 16.81 | 11.63 |
| | 72B-GUI-Judge | 21.55 | 14.13 |
| | 72B-Gudge-Ensemble | 30.17 | 19.39 |
| 72B-Gudge-Ensemble BC (SFT) | | 28.02 | 10.25 |

ward provider paradoxically degrades the Success Rate (SR) after training. In contrast, our finetuned judge model can achieve a performance improvement, which is further enhanced by the ensemble voting strategy proposed in Sec. 4.2. To illustrate this, we visualize the training process under different reward signals in Fig. 5. The visualization confirms that only the reward from our proposed ensembled judge system leads to a improving training reward. This finding aligns with the emphasis on reward quality highlighted by Lu et al. (2025); Wang et al. (2025); however, our approach achieves this training reward trajectory automatically with strong reward system, whereas Lu et al. (2025) relied on a human-labeled verification function within OSWorld (Xie et al., 2024).

To evaluate domain generalizability, we test our model on a cross-domain task, training it exclusively on the ScienceBoard suite and evaluating it directly on the OSWorld benchmark (Xie et al., 2024). As reported in Tab. 3 and visualized in Fig. 5 (b), our decoupled reinforcement learning approach demonstrates strong generalization, achieving a significant improvement in success rate on the unseen domain. In sharp contrast, the behavior cloning (SFT) baseline exhibits poor generaliz-

ability, suffering a substantial performance drop when transferred to the new software environment. These findings suggest that our RL approach learns a more robust and generalizable policy, whereas SFT is prone to overfitting to the superficial patterns of the training environment.

We further analyze the effectiveness and data efficiency of our decoupled reinforcement learning framework, as proposed in Sec. 3.3.1. As illustrated in Fig. 5 (c), conventional direct reinforcement learning on powerful models such as Qwen2.5-VL-32B (Bai et al., 2025) and UI-TARS-1.5-7B (Qin et al., 2025) yields only marginal improvements in success rate. In contrast, our decoupled approach achieves substantially better performance in significantly fewer training steps. This is accomplished by keeping the UI-TARS-1.5-7B model fixed to generate final actions while focusing the optimization exclusively on the planner by applying the advantage signal to its planning tokens. This result validates our decoupled design, showing it is a more effective and data-efficient strategy for software adaptation, as it allows domain-specific knowledge to be learned and localized entirely within the planner.

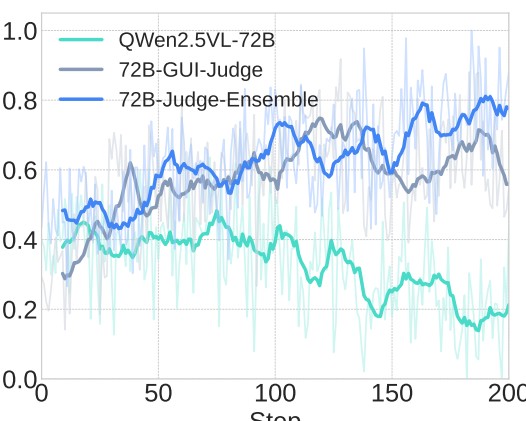

Figure 5: **Training Reward** with different judge model during training process. Training with our judge system can lead to increase in training reward compared to other judge baseline.

**Generalizability to Heterogeneous Executors.** From a tool-use perspective, the executor (e.g., GUI-Actor (Wu et al., 2025)) can be viewed as an external tool that the planner learns to call. To rigorously test the generalizability of our decoupled framework, we investigate whether the planner, trained exclusively with a GUI-based executor, can adapt to a fundamentally different tool type: a command-line

Table 4: Code agent as additional executor for evaluation of cross executor generalizability.

| Methods | Biochem (%) | Algebra (%) |
|---|---|---|
| CODA (Qwen2.5VL-32B) | 44.83 | 41.94 |
| Qwen2.5-72B | 41.38 | 35.48 |
| Qwen2.5-72B w/ CODA | 55.17 | 48.39 |
| Qwen3-32B | 44.83 | 38.71 |
| Qwen3-32B w/ CODA | 58.62 | 51.61 |

coder agent. We achieve this by simply augmenting the planner's action space with a `cmd` command, which delegates execution to an external coder model (see prompt details in Fig. 14). The results, presented in Tab. 4, show compelling evidence of cross-tool generalization. When powerful base models like Qwen2.5-72B (Wang et al., 2024) and Qwen3-32B (Yang et al., 2025) are used directly as code agents, they establish a strong performance baseline. However, when these same models are guided by our CODA-trained planner, their success rates increase substantially across both benchmarks. For instance, on the Algebra software, applying CODA boosts the performance of composing code agent from 35.48% to 48.39%. This is a significant finding: the planner was never trained with the code agent, yet it effectively leverages it to solve tasks. This suggests that our decoupled reinforcement learning approach enables the planner to learn more than just software-specific domain knowledge; it learns a more abstract, generalizable policy of how to call and utilize an external tool. This learned skill transfers seamlessly from a grounding executor to a code agent, underscoring the flexibility and extensibility of our proposed framework.

## 5 CONCLUSION

We presented a trainable Planner–Executor disentangled framework for GUI agents, inspired by the division of labor between the cerebrum and cerebellum. By coupling a fixed executor with a fine-tunable planner, and supporting it with a robust judging system, GRPO-based exploration, and a distributed data generation pipeline, our approach effectively addresses the challenges of complex interfaces and long-horizon planning. Experiments on ScienceBoard demonstrate substantial improvements over strong baselines with strong generalizability to out-of-domain software and novel executor (code agent). These results highlight the importance of combining stable execution with adaptive planning, and open promising directions for extending our framework to richer multi-modal feedback, broader professional domains, and continual learning for long-term adaptability.

ETHICS STATEMENT

Our work adheres to the ICLR Code of Ethics. Our research, which focuses on creating autonomous agents for complex scientific and engineering software, is intended for beneficial applications such as accelerating scientific discovery, while we acknowledge that the underlying technology could be misused. To mitigate this, our experiments are conducted exclusively within controlled virtual machine environments. The foundation models used may inherit societal biases, and we recognize our automated judging system could potentially learn similar biases. A key ethical contribution of our work is the development of a training paradigm that eliminates the need for costly human-labeled trajectories, reducing the reliance on manual annotation labor. Furthermore, our decoupled framework directly addresses the environmental impact of training large models by offering a more data- and compute-efficient approach than end-to-end training. We are committed to the responsible advancement of transparent and controllable autonomous agents.

REPRODUCIBILITY STATEMENT

We are committed to full transparency and reproducibility. Following the peer-review process, we will open-source our complete project. This includes **all source code** for our decoupled reinforcement learning framework, the automated judging system, and all training and evaluation scripts, as well as **all model weights** for our fine-tuned planner agents and judging system. Our framework is built upon publicly available base models (Qwen2.5-VL (Bai et al., 2025), UI-Tars-1.5 (Qin et al., 2025) and GUI-Actor (Wu et al., 2025)), for which we provide exact identifiers. The appendix further provides a comprehensive guide to the experimental setup, detailing software configurations for the ScienceBoard and OSWorld benchmarks, all training hyperparameters, and the computational resources required to fully replicate our findings.

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

## LLM Usage Disclosure

We leveraged Large Language Models (LLMs), such as Google's Gemini, to aid in the preparation and polishing of this manuscript. The use of these models was confined to the role of a sophisticated proofreading tool. Specifically, we used the LLM for tasks including correcting grammatical errors, refining sentence structure for better clarity, and improving the overall readability of the text.

It is crucial to state that all content, including the research ideation, experimental design, data analysis, and conclusions, was generated entirely by the human authors. Every suggestion from the LLM was critically evaluated, and the authors performed multiple rounds of review and revision to take full responsibility for the final version of the text. The LLM was not used to generate any core scientific insights or substantive parts of the paper.

## A  Case Study

We provide case study in Fig. 6 where Qwen2.5-VL-72B (Bai et al., 2025) struggles with precise grounding, while UI-TARS-1.5 (Qin et al., 2025), though specialized, fails to generalize to out-of-distribution software.

## B  Judge Model Fine-tuning Details

Inspired by (Sun et al., 2025b), we adopt a fine-tuning approach to obtain a strong judge model. We scale up the model to Qwen2.5-VL-72B (Bai et al., 2025), and use a dataset comprising 4.7K labeled judgment samples. These trajectories are generated by Qwen2.5-VL and Gemini-2.0-Pro on WebArena (Zhou et al., 2023), UI-TARS-1.5 (Qin et al., 2025), and GPT-4o (OpenAI, 2023) on OSWorld (Qin et al., 2025). Judgments are provided by GPT-4o and Gemini2.5-Pro (Google DeepMind, 2025), with detailed captions for each screenshot frame during agent execution. The judgments are further filtered, retaining only those that align with verified ground-truth results. Additionally, change description data is incorporated inspired by SEAgent (Sun et al., 2025b).

Training is conducted on 32 A100 GPUs for 370 steps, using LoRA (Hu et al., 2022) with a rank of 8. The resulting model, trained on OSWorld trajectories, generalizes well to AgentRewardBench (Lù et al., 2025) and ScienceBoard (Sun et al., 2025a). This fine-tuned model is referred to as **72B-GUI-Judge** in Table 2, and demonstrates improved precision on two out-of-domain benchmarks. When further ensembled with the original 72B base model, it achieves even higher precision, providing more accurate reward signals—crucial for effective reinforcement learning of the planner agent.

## C  Training data statistics.

Table 5: Training data statistics on different softwares "#trajectories (#multi-turn conversation pairs)".

| Stage | Algebra | Biochem | GIS | Astron |
|---|---|---|---|---|
| Stage1 | 215 (1193) | 199 (1205) | 273 (1433) | 165 (975) |
| Stage2 | 314 (1587) | 224 (1271) | 351 (1688) | 221 (1104) |

For both training stage, we use prompt detailed in Fig. 12 to leverage QWen2.5-72B (Wang et al., 2024) for generating high level tasks. We generate 1000 task for each software and ask corresponding agent to generate 8 times to get 8000 trajectories, these trajectories are labeled by our judging system, leaving around 300 positive trajectories for each software. During second stage, these data are seen as 5.6K multi-turn conversation pairs for aggregated supervised fine-tuning.

**Qwen2.5-VL-72B**

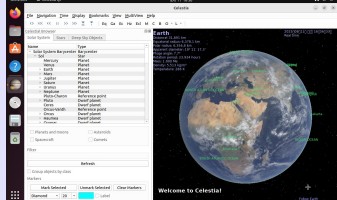 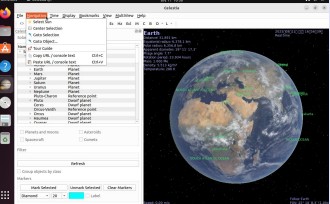 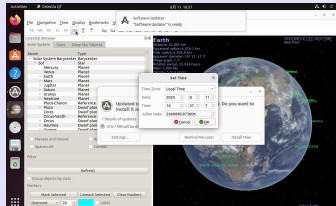

To set the Julian date in Celestia to 2400000, we need to access the Time menu. From the screenshot, I see that the \"Time\" menu is located in the top menu bar. The next step is to click on the \"Time\" menu to find the option for setting the Julian date.\n\nHere's the code to click on the \"Time\" menu: python\npyautogui.click(x=188, y=73)

It seems I clicked the wrong menu. The task requires setting the Julian date, so we need to access the \"Time\" menu instead of the \"Navigation\" menu. Let's correct this by clicking on the \"Time\" menu, which is adjacent to \"Navigation\".\n\nHere is the corrected code to click on the \"Time\" menu:\n\n```python\npyautogui.click(x=213, y=73)

A pop-up notification from the Software Updater has appeared, which may obstruct our view of the \"Set Time\" dialog box. Before proceeding to enter the Julian date, it's important to dismiss or move the pop-up out of the way to fully interact with the dialog\n\nHere is the code to close the Software Updater pop-up: python\npyautogui.click(x=922, y=55)

**UI-TARS-1.5-7B**

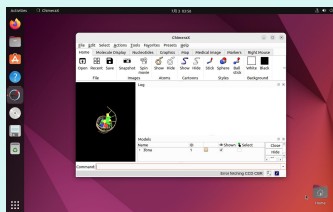 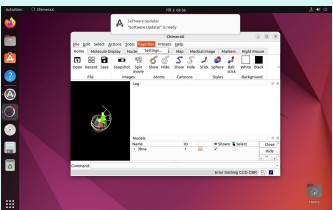 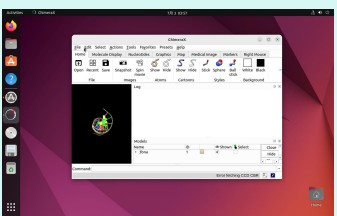

Thought: Let me take a look at the interface layout. I notice there's a \"Favorites\" option in the top toolbar, which seems to be the entry point I need. I need to click on it to expand the menu and see if there are any useful options available to help me with the annotation work.\nAction: click(start_box='(554,148)')

Thought: I took a close look at the current interface and realized that I had been heading in the wrong direction before. I now need to get back on track by clicking on the \"Tools\" option in the top menu bar. This will allow me to find the functionality for adding labels and notes.\nAction: click(start_box='(505,149)')

Thought: I've been looking at the interface, and it's clear that just clicking on the tools menu doesn't provide the feature I need for adding labels and notes. It seems I need to approach this differently. There's a Home option in the top menu bar, and I should be able to find the tools I'm looking for there..\nAction: click(start_box='(314,149)')

Figure 6: **Case studies.** Qwen2.5-VL-72B (Bai et al., 2025) struggles with precise grounding, whereas UI-TARS-1.5 (Qin et al., 2025), though specialized, fails to generalize to out-of-distribution software.

## D PROMPT DETAILS.

We provide detailed prompt for task generator in Fig.12 and judge system in Fig.13. Detailed prompt for planner agent is in Fig.14. Prompt we used for executor agent aligns with UI-TARS (Qin et al., 2025) official code.

## E VIRTUAL MACHINE SYSTEM DETAILS

We utilized a local cluster consisting of 15 servers to collect interaction trajectories. Among these, 13 servers were equipped with AMD EPYC 7742 processors, and 2 servers were equipped with Intel i9-13900K CPUs paired with NVIDIA GeForce RTX 4090 GPUs to support software with

**Task Instruction: Go to the brightest star in absolute magnitude within 20ly of Earth in Celestia.**

## CODA (Ours)

**Thought:** The brightest star in absolute magnitude within 20 light years of Earth should be Sirius. I will select the 'Stars' tab in the Celestial Browser to find Sirius.
**Action:** click(start_box='(204,161)')

**Thought:** In the current list of stars within 20 light years from Earth, Sirius A has an absolute magnitude of -1.46 which makes it one of the brightest. Therefore, selecting Sirius A would take us to this star in Celestia.
**Action:** click(start_box='(129,410)')

**Thought:** The context menu for Sirius A is open, and the 'Goto' option is visible. Selecting this will navigate Celestia to Sirius A.
**Action:** click(start_box='(143,428)')

**Thought:** Since Sirius A is already thxe selected and traveled destination, no further actions are necessary to achieve the goal.
**Action:** wait()

**Thought:** Given that Sirius A is both selected and traveled to, and its details are displayed in the celestial browser, the task to visit the brightest star in absolute magnitude within 20 light years of Earth is completed successfully.
**Action:** finished()

## Qwen2.5–VL–72B

**Thought:** To navigate to the brightest star in absolute magnitude within 20 light-years (ly) of Earth in Celestia, we should start by switching to the \"Stars\" tab in the Celestial Browser.
**Action:** click(start_box='(208,161)')

**Thought:** let's check the list for stars with the lowest absolute magnitude. From the list visible, Sirius A has an absolute magnitude of -1.46, making it the brightest within this range. Now, let's select Sirius A to focus on it. Click on Sirius A in the list. If necessary, confirm the selection to view it in the main display.
**Action:** click(start_box='(150,415)')

**Thought:** The task is to navigate to the brightest star in absolute magnitude within 20 light-years of Earth. The list shows several stars with their distances and absolute magnitudes. Based on the list:\n\n- Sirius A has an absolute magnitude of -1.46, which is the brightest among the listed stars.\n\nHowever, we previously clicked on BL Cet, which may have been unintended. Let's ensure we select Sirius A instead.
**Action:** click(start_box='(149,414)')

**Thought:** Sirius A has already been identified as the brightest star within 20 light-years of Earth, with an absolute magnitude of -1.46. However, the star currently selected is BL Cet, which has an absolute magnitude of 15.56. To proceed with the task, we need to reselect Sirius A to focus on it in Celestia.
**Action:** click(start_box='(150,462)')

**Thought:** The current selection in the Celestial Browser is still BL Cet, despite our intention to focus on Sirius A, which has the brightest absolute magnitude (-1.46) within 20 light-years of Earth. To proceed with the task, we must ensure Sirius A is selected instead.
**Action:** click(start_box='(150,451)')

## UI–TARS–1.5–7B

**Thought:** I took a quick glance at the star list, and I noticed that Sirius A is the closest bright star to Earth, with an absolute magnitude of 1.43. However, I need to check if it's within a 20-light-year radius. I clicked on the \"Closest Stars\" option to filter the results, which will help me confirm whether this star meets all the criteria.
**Action:** click(start_box='(158,532)')

**Thought:** I just clicked on the \"Closest Stars\" option, but it seems like the system didn't respond. Let me check the status of the interface – the \"Closest Stars\" option is still selected. It looks like I need to click that option again to see if it works. After all, in order to find the star with the lowest absolute magnitude, I need to ensure that the system is correctly filtering the stars that are closest to Earth.
**Action:** click(start_box='(158,532)')

**Thought:** I took a look at the star list, and the closest star to Earth is indeed Proxima Centauri, which is 4.246 light-years away. However, I need to find the star that is not only the closest but also the brightest. So, I'll switch to the \"Brightest Stars\" option to see what I can find. This way, I can accurately identify the star that has the lowest absolute magnitude within a 20-light-year radius.
**Action:** click(start_box='(327,532)')

**Thought:** I took a look at the current list of stars, and I found that the Sun is the one with the lowest absolute magnitude among the closest stars to Earth. However, the task requires me to find the star with the lowest absolute magnitude within a 20-light-year radius, not just the closest one. So, I need to keep browsing through the list to see if there are any other stars that meet the criteria.
**Action:** scroll(direction='down', start_box='(327,385)')

**Thought:** I've been scrolling through the star list for quite a while now, but I still haven't come across the star with the lowest absolute magnitude. It seems that I need to keep looking further down. Although the current list already shows several stars with lower absolute magnitudes, I believe that there might be even brighter stars waiting for me to discover. So, I'll continue to scroll down to see what other surprises the list has in store for me.
**Action:** scroll(direction='down', start_box='(327,385)')

Figure 7: **Full trajectory case studies.** Qwen2.5-VL-72B (Bai et al., 2025) struggles with precise grounding, whereas UI-TARS-1.5 (Qin et al., 2025), though specialized, fails to generalize to out-of-distribution software. In this case, Qwen straggle to precisely select "Serius A" while UI-TARS having trouble choosing the correct star.

high graphical computing demands, such as ChimeraX. Using VMware Workstation Pro, we ran 4 to 8 independent virtual machines concurrently on each server to execute tasks in parallel.

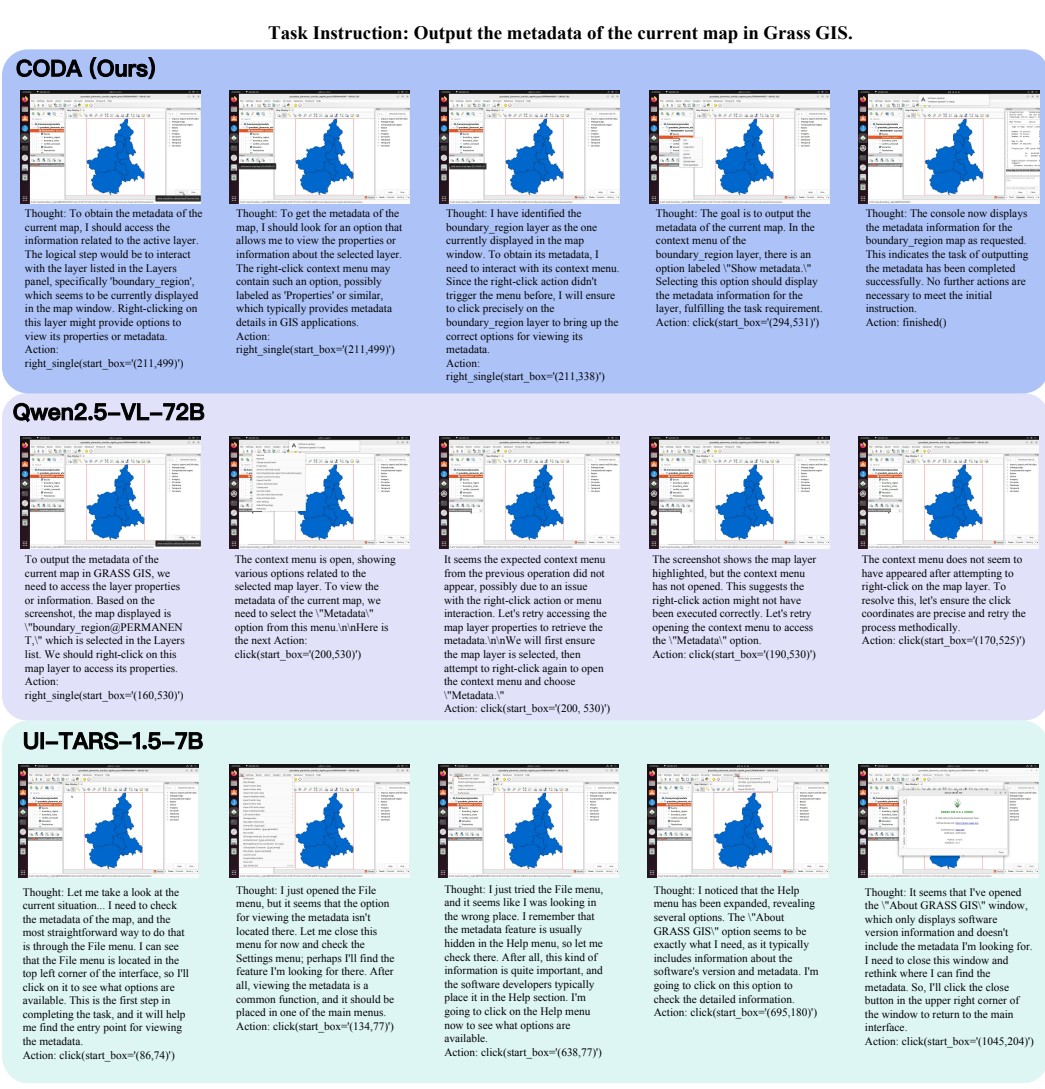

Figure 8: **Full trajectory case studies.** Qwen2.5-VL-72B (Bai et al., 2025) struggles with precise grounding, whereas UI-TARS-1.5 (Qin et al., 2025), though specialized, fails to generalize to out-of-distribution software. In this case Qwen straggle to precisely click "metadata" item. UI-TARS does not contain the domain knowledge to perform the task and tries to get help.

**300 step**

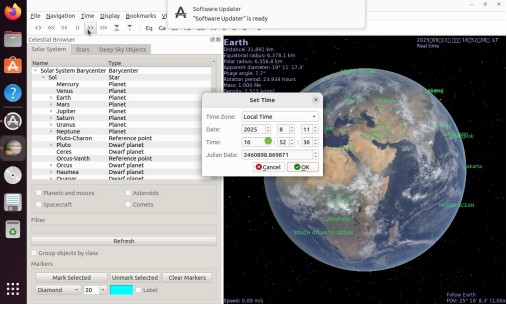

**100 step**

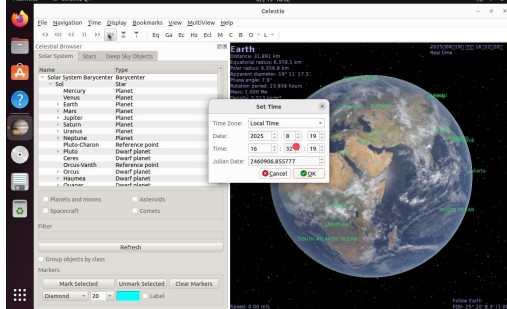

**Thought:** I need to set Celestia's current time to 3 hours later. I see the current time is 16:52. I will click the small upward triangle button for the hour to increase the time. Because the button is very small, I am going to call the grounding model. GROUND_CLICK(\"click the small upward triangle button for the 16 hour box\")
GROUND_CALL_RETURN(click(start_box='(681,385)'))
**Action:** click(start_box='(681,385)')

**Thought:** To set Celestia's current time to 3 hours later, I need to click the button for setting the hour in the Time [window]. I see the current time is 16:32, and I will call the grounding model to click the small upward triangle button to increase the time.
GROUND_CLICK(\"the small triangle button to increase time\")
GROUND_CALL_RETURN(click(start_box='(736,387)'))
**Action:** click(start_box='(736,387)')

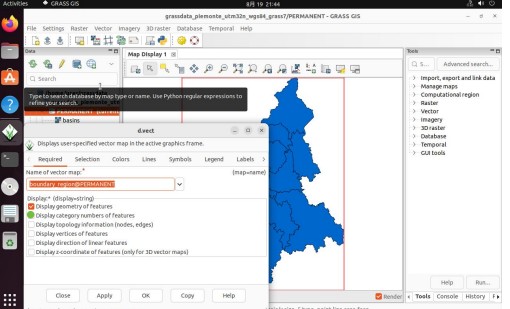

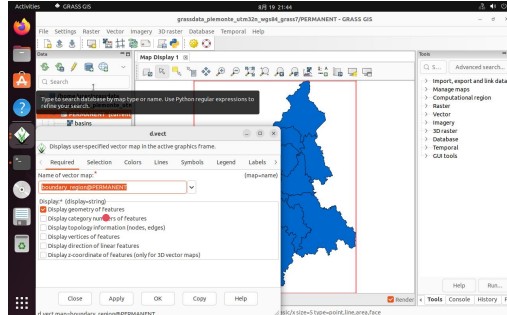

**Thought:** To display the category numbers and topology information, I need to check the corresponding checkboxes in the \"d.vect\" dialog. The options for \"Display category numbers of features\" and \"Display topology information (nodes, edges)\" are visible and need to be enabled.
GROUND_CLICK(\"checkbox of Display category numbers of features\")
GROUND_CALL_RETURN(click(start_box='(88,548)'))
**Action:** click(start_box='(88,548)')

**Thought:** To output the category number and topology information, I need to enable the corresponding checkboxes in the \"d.vect\" dialog. The options for \"Display category numbers of features\" and \"Display topology information (nodes, edges)\" are currently unchecked.
GROUND_CLICK(\"Display category numbers of features\")
GROUND_CALL_RETURN(click(start_box='(230,549)'))
**Action:** click(start_box='(230,549)')

Figure 9: **Case studies of Grounding tool call.** The left column displays the further trained model (grounding positions marked with green dots), while the right column represents the early training stage (grounding positions marked with red dots). At the early stage, the planner failed to call the grounding tool with precise instructions, leading to confusion. After more steps of RL, the planner learns to generate more detailed instructions for precise grounding.

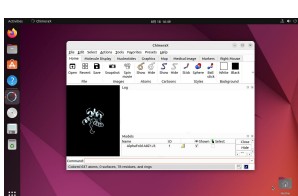 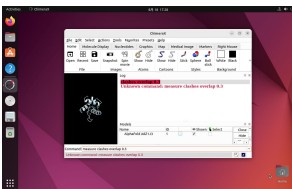 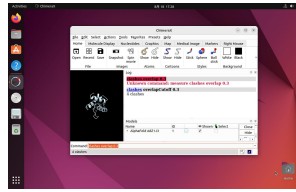

**300 step**

**Task Instruction**

Detect all clashes with VDW overlap ≥ 0.3Å in ChimeraX.

**Thought:** To detect clashes with VDW overlap of at least 0.3 Å in ChimeraX, we need to use the `clashes` command. This command allows us to specify the threshold for detecting clashes. Since the Command Line Interface (CLI) is visible, we can directly input the command there.
**Action:** type(content='clashes overlap 0.3')

**Thought:** My last attempt to run the command myself failed, returning 'unknown command'. I am switching to use the code model to help me write this command.
CMD(\"Generate a single command to \"Detect clashes with van der Waals (VDW) overlap of at least 0.3 Å in ChimeraX\"\")
CMD_RETURN ('clashes overlapCutoff 0.3')
**Action:** type(content='clashes overlapCutoff 0.3')

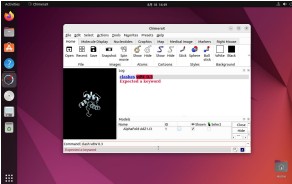 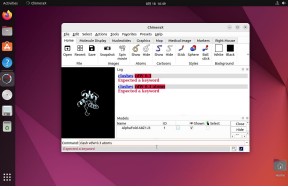 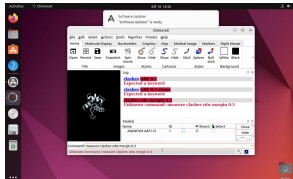

**100 step**

**Thought:** To detect clashes with VDW overlap of at least 0.3 Å in ChimeraX, we need to use the `clashes` command. This command allows us to specify the threshold for detecting clashes.
**Action:** type(content='clashes vdw 0.3')

**Thought:** My last attempt at the command failed, returning 'Expected a keyword'. I will use the code model to help me write this command.
CMD(\"clashes vdw 0.3\")
CMD_RETURN ('clashes vdm 0.3 atoms')
**Action:** type(content='clashes vdm 0.3 atoms')

**Thought:** My last attempt at the command failed, reporting error 'Expected a keyword'. I am going to call the code model to help me write this command.
CMD(\"clashes vdm 0.3 atoms\")
CMD_RETURN ('clashes vdm margin 0.3 atoms')
I need to type this command into CLI.
**Action:** type(content='clashes vdm margin 0.3 atoms')

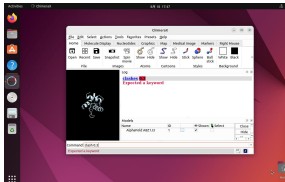 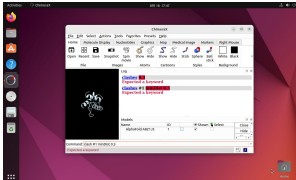 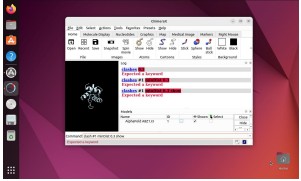

**Before Training**

**Thought:** To detect clashes with van der Waals (VDW) overlap ≥ 0.3Å in ChimeraX, we will utilize the command-line interface. The goal is to identify and visualize overlapping atoms that are too close to each other, exceeding the standard VDW radius threshold. Execute the `clash` command.
**Action:** type(content='clashes 0.3')

**Thought:** The `clash` command was attempted with the parameter `0.3`, but an error message indicates that ChimeraX expected a keyword after the `clash` command. we can explicitly specify the model and adjust the threshold to detect clashes effectively `minDist 0.3` sets the minimum distance threshold for detecting clashes to 0.3 Å
**Action:** type(content='clashes #1 minDist 0.3')

**Thought:** The `clash` command is being executed incorrectly. Despite the attempt to specify `#1 minDist 0.3`, ChimeraX still reports an error indicating that it \"Expected a keyword.\" A common approach is to use the `show` option to visualize clashes directly.
**Action:** type(content='clashes #1 minDist 0.3 show')

Figure 10: **Case studies illustrating the generalization of code tool usage at different training steps. Bottom Row (Before Training):** Although the action of CMD is written in the system prompt, the planner failed to identify the appropriate moment to invoke it. Instead, in this example, it repeatedly relied on hallucinated internal knowledge, resulting in a sequence of incorrect CLI commands without utilizing the external code agent. **Middle Row (100 Steps):** The model begins to demonstrate strategic invocation, recognizing its internal failure and switching to the CMD tool to rectify the error. **Top Row (300 Steps):** Driven by shared structural patterns with grounding tools, the planner generates significantly more informative and precise instructions for the code agent (e.g., a detailed natural language description). This confirms a general enhancement in reliable, structured tool usage, where the planner learns to leverage external tools specifically when internal generation fails.

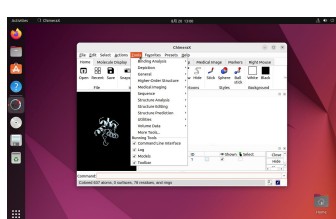

**Thought:** To detect clashes, I need to use the \"Tools\" menu, which typically contains options for analyzing molecular structures. The \"Tools\" menu is visible in the toolbar at the top of the ChimeraX window. Selecting this menu should provide access to clash detection features. I need to call the grounding model as my click is inaccurate.
GROUND_CLICK(\"click the \"Tools\" menu in the toolbar\")
GROUND_CALL_RETURN(start_box='(511,149)'))
**Action:** click(start_box='(511,149)')

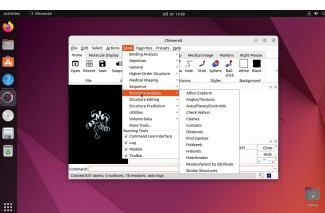

**Thought:** The \"Structure Analysis\" option under the \"Tools\" menu is likely to contain features for detecting atomic clashes based on Van der Waals (VDW) overlaps. This is a common feature in molecular visualization software for identifying steric hindrance.
**Action:** click(start_box='(549,323)')

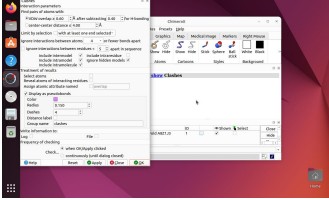

**Thought:** The \"Clashes\" option under \"Structure Analysis\" is the relevant tool for detecting atomic clashes. Selecting this option will likely open a dialog or panel where I can set the threshold for VDW overlap detection.
**Action:** click(start_box='(752,425)')

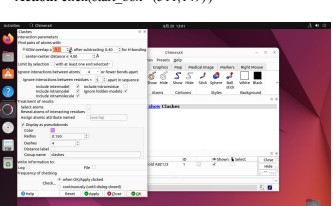

**Thought:** The current VDW overlap threshold is set to 0.60 Å, but the task requires detecting clashes with a VDW overlap ≥ 0.3 Å. I need to adjust the threshold to 0.3 Å and then apply the changes to detect the correct clashes. Because the input box is small, I need to switch to the grounding model.
GROUND_CLICK(\"click input box of VDW overlap currently showing 0.60 Å\")
GROUND_CALL_RETURN(click(start_box='(269,104)'))
**Action:** click(start_box='(269,104)')

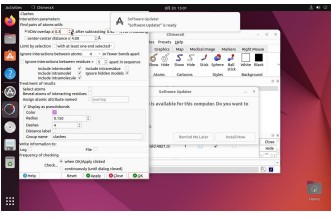

**Thought:** The VDW overlap field is currently set to 0.70 Å. I need to change this value to 0.3 Å to meet the task requirement. After updating the field, I will apply the changes to detect clashes accurately.
**Action:** type(content='0.3')

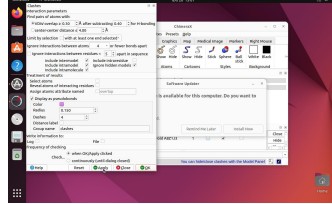

**Thought:** The VDW overlap threshold has been correctly set to 0.3 Å. To apply these settings and detect clashes, I need to click the \"Apply\" button. This will refresh the clash detection based on the updated criteria. I will call the grounding model.
GROUND_CLICK(\"click the \"Apply\" button.\")
GROUND_CALL_RETURN(click(start_box='(366,673)'))
**Action:** click(start_box='(366,673)')

Figure 11: **Case study of inefficient pure GUI approach to complete the task.** Compare to single command line in Fig. 10, pure GUI operation require 6 steps to complete the same task.

## Task Buffer Update Prompt Template

You are an expert task designer for autonomous AI agents. Your role is to generate a comprehensive and diverse set of tasks for a Computer Use Agent (CUA) that will be trained to operate a specific software application.

The generated tasks will serve as high-level objectives within a two-stage training process: 1) **decoupled reinforcement learning**, where the agent explores the software, and 2) **aggregated fine-tuning** on successful interaction trajectories.

### **Target Software**

`{software_name}`: {software_description}`

### **Task Generation Guidelines**

1.  **High-Level Objectives**: Tasks must describe *what* to achieve, not the specific steps on *how* to achieve it. They must be goal-oriented. For example, "Create a pivot table summarizing sales by quarter" instead of "Click on cell A1, then go to the Insert tab...".
2.  **Diversity and Coverage**: The tasks must cover a wide range of the software's functionalities, from basic, single-step operations to complex, multi-step workflows that combine different features.
3.  **Clarity**: Each task must be a clear, unambiguous instruction that can be evaluated for success.

### **Your Instruction**

Based on the software description above, generate a list of {number_of_tasks} distinct tasks.

### **Output Format**

Your response must be **only** a single JSON object containing a list of task strings. Do not include markdown formatting (`json`), reasoning, or any other explanatory text in the output.

```json
[
    "task1",
    "task2",
    "...",
    "task50",
]
```

Figure 12: **Detailed prompt for task generation.**

1188
1189
1190
1191
1192
1193
1194
1195
1196
1197
1198
1199
1200
1201
1202
1203
1204
1205
1206
1207
1208
1209
1210
1211
1212
1213
1214
1215
1216
1217
1218
1219
1220
1221
1222
1223
1224
1225
1226
1227
1228
1229
1230
1231
1232
1233
1234
1235
1236
1237
1238
1239
1240
1241

## Judge Prompt Template

I am evaluating the performance of a UI agent. The images provided are **sequential keyframes** that represent
the full execution trajectory of the agent when attempting to follow a command.
These keyframes correspond to the instruction: **'{instruction}'**.

Please thoroughly analyze the sequence to assess the following aspects:
1. **Correctness** — Did the agent successfully complete the task as instructed?
2. **Redundant Steps** — Identify any unnecessary or repeated actions that do not contribute to the goal.
3. **Optimization** — Did the agent follow an efficient plan with a minimal number of steps?
4. **First Error Step** — If the execution is incorrect or sub-optimal, determine the index of the **first keyframe where a mistake occurred**.
5. **Error Analysis** — Provide a brief explanation of the mistake at that step.
6. **Correct Action Suggestion** — Explain what the agent **should have done instead** at the point of error.
----------------------------------------------------------------------------------------------------
**Important Instructions:**
- The agent may have made progress toward the goal, but unless the task is **fully and correctly completed**, you must set 'Correctness' to **False**.
- Be cautious in determining success. Missing confirmation screens, skipped inputs, or wrong UI elements clicked all count as errors.
- Carefully examine all UI changes, button interactions, text entries, and any visual feedback in the screenshots.
- Clearly indicate **which exact steps are redundant** (starting from 1).
----------------------------------------------------------------------------------------------------
You output must be extremly concise and focused, with clear emphasis on key points. If the agent fails, only provide the core reason for the first step failure, ignoring other minor issues. Keep the language clear and direct.

This is a Question-Answering Task. In addition to the action sequence, the agent's final goal is to provide a correct answer.
The agent submitted the final answer: '{agent_answer}'
Your evaluation must also determine if this answer is correct based on the information visible in the **final screenshot**
----------------------------------------------------------------------------------------------------
Once you finish the analysis, return your evaluation in the following dictionary JSON format:

<captions>
Frame1: caption of the first frame
Frame2: caption of the second frame
(max 15 frame captions)
</captions>
<res_dict>{
  "Correctness": True/False,
  "Redundant": [step_num, ...],
  "Optimized": True/False,
  "First_Error_Step": step_num or None,
  "Error_Type": "brief description of the mistake",
  "Correct_Action": "what should have been done instead"
}</res_dict>

Figure 13: **Detailed prompt for the judge model.** Text in gray all task type based on whether finish given task or answer the question from user.

**Planner Prompt Template on ScienceBoard**

You are a GUI agent. You are given a task and your action history, with screenshots. You need to perform the next action to complete the task.
## Output Format
```

Thought: ...
Action: ...
```

- Write a small plan and finally summarize your next action (with its target element) in one concise sentence in `Thought` part.
## Action Space
**++click('description of the button/icon to be click') # calling grounding model.**
left_double('description of the button/icon to be click')
right_single('description of the button/icon to be click')
drag(start_box='(x1,y1)', end_box='(x3,y3)')
hotkey(key='')
type(content='') #If you want to submit your input, use "\\n" at the end of `content`.
**++cmd('description of the command') # calling coder model.**
scroll(start_box='(x1,y1)', direction='down or up or right or left')
wait() #Sleep for 5s and take a screenshot to check for any changes.
finished()
call_user() # Submit the task and call the user when the task is unsolvable, or when you need the user's help.
**answer('your answer to user question')**
## User Instruction
when generating action, generate action only without any comments or other text.
strictly follow the output format. And generate only ONE action listed above with thinking process.
{instruction}

Figure 14: **Detailed prompt for the planner agent.**

