# OpenReview forum: "CODA: Coordinating the Cerebrum and Cerebellum for a Dual-Brain Computer Use Agent with Decoupled Reinforcement Learning"
_ICLR.cc/2026/Conference — Submitted to ICLR 2026_

### Official Review · Reviewer_JvNe · 2025-10-30

**Soundness:** 2
**Presentation:** 2
**Contribution:** 2
**Rating:** 4
**Confidence:** 5

**Summary:**

This paper introduces CODA, a dual-module (planner-executor) framework for GUI agents inspired by human brain anatomy. The core idea is to separate a "cerebrum" (Planner) responsible for high-level reasoning from a "cerebellum" (Executor) that handles precise action execution. The method's novelty lies in its two-stage training paradigm: (1) **software-specific specialist** planners are first trained via decoupled reinforcement learning (with a static executor); (2) successful trajectories from all specialists are then aggregated for **supervised fine-tuning (SFT)** of a generalist planner. To avoid manual annotation, the paper details an automated "Judge System" to provide reward signals and presents a scalable distributed infrastructure for data collection. Experiments on ScienceBoard and OSWorld demonstrate improvements over baselines and claim robust generalization to unseen software and, surprisingly, heterogeneous executors like code agents.

**Strengths:**

1. Principled Decoupling: The cerebrum-cerebellum analogy is intuitive, but more importantly, it offers a practical and modular solution to the long-standing tension between high-level planning and fine-grained execution. This design allows for efficient adaptation to new software without retraining the entire agent.
2. Technically Rigorous Two-Stage Training: The "specialist-to-generalist" curriculum (RL -> SFT) is well-motivated, directly addressing the limitations of monolithic, end-to-end training. The ablation studies and cross-domain results (Tables 3, 4) provide strong empirical support for this choice.
3. Automated Judging System: The judging system provides scalable reward signals via ensemble voting and multi-resolution input (Table 2), which significantly reduces the dependency on manually labeled trajectories, a strong point for real-world applicability.
4. Solid and Consistent Empirical Gains: CODA achieves new SOTA performance for open-source models on ScienceBoard, with consistent improvements shown across multiple software domains and metrics.
5. Commitment to Open Science: The authors' pledge to release all code and models is excellent for the community and reproducibility.

**Weaknesses:**

Despite the strengths, the paper fails to convince me on several critical points:

1. Lack of Critical Baselines; SOTA Claims Unfounded:
   - This is the paper's most severe flaw. The baseline comparison in Table 1 is missing the most crucial and direct competitor: a top-tier closed-source planner (like GPT-4o) paired with the *same* open-source executor (like UI-TARS). This is the "gold standard" for evaluating the CODA planner's performance. Without it, the SOTA claim is unsubstantiated.
   - Furthermore, as listed in "Potentially Missing Related Work," a large body of work on "decoupled architectures" and "hierarchical RL" already exists. The paper makes no empirical comparison to these structurally similar open-source works (e.g., Wang et al. 2024, Chen et al. 2023), clouding CODA's true innovative contribution.
2. Significant Concerns about the Automated Judge System:
   - Strange Ensemble Architecture: The Judge's design is perplexing. The authors claim to "distill" knowledge from GPT-4o/Gemini into the 72B-GUI-Judge, but then must ensemble this "student" model with the original pre-trained model for the final system. This seems to be a vote of no confidence in the distillation itself. If the fine-tuning was successful, why is the original model still needed?
   - Lack of Teacher Benchmarks: The paper never reports the accuracy (Precision/Recall) of the "teacher" models (GPT-4o/Gemini) on the judging tasks. We have no way to know the quality of the distillation data source or to evaluate how well the student model learned.
   - Low Metrics and Potential for Bias: The reported Precision/Recall numbers in Table 2 (many below 80%) are worrying. Low precision introduces false positive rewards, which can destabilize RL training and teach "hallucinated" behaviors. Low recall discards valuable successful trajectories, hurting the data efficiency of the SFT stage.
   - Why not use simpler, more robust methods? For many tasks, determining "success" could be achieved with near-100% accuracy and without model-induced bias. This could be done by training a task-specific model, using rule-based oracles, or leveraging other oracle information. Why did the authors opt for this complex, lower-accuracy, and potentially biased distillation approach instead of these simpler, more reliable methods?
3. Insufficient Evidence for Cross-Executor Generalization:
   - This is one of the paper's most surprising claims (Table 4), but it is the least explained. Why would a planner that has only ever been trained to command a GUI executor suddenly understand how to command a code agent it has never seen?
   - This generalization is counter-intuitive. It implies the planner learns a highly abstract, executor-agnostic policy ("Thought"). If true, this is a major finding, but the paper provides zero case studies to analyze this. We need to see a side-by-side comparison of the "Thought" process as CODA commands a GUI agent versus a code agent for the same task.
4. Unfair and Incomplete Case Studies:
   - The case study in Figure 6 is not a convincing comparison. It shows Qwen2.5-VL failing on Case A and UI-TARS failing on Case B. This is a classic "cherry-picking" setup.
   - A meaningful comparison must test all models (Qwen, UI-TARS, CODA) on the exact same cases. How does UI-TARS perform on Case A? How does Qwen perform on Case B? How does CODA perform on both? The current analysis is uninformative.
5. The Static Executor: A Potential Bottleneck:
   - The CODA framework assumes the "cerebellum" (executor) is perfect and static. This is a risky assumption for real-world applications. If faced with a truly novel, out-of-distribution (OOD) GUI layout or widget, the fixed executor will fail, and the planner has no recourse.
   - The paper never discusses this "execution bottleneck" or the conditions under which a static executor would become the system's limiting factor.

**Questions:**

1. **(Re: Baselines)** Can the authors provide results for the single most important baseline: **GPT-4o (or Gemini 2.5 Pro) as the Planner** paired with the UI-TARS-1.5 Executor?
2. **(Re: Judge System)**
   - Please justify the "distilled model + pre-trained model" ensemble. Why is this necessary if the distillation was successful?
   - What is the Precision/Recall of the "teacher" models (GPT-4o/Gemini) on your judging tasks?
   - Why not use a task-specific model or a rule-based oracle for rewards to achieve higher accuracy and avoid model bias?
3. **(Re: Cross-Executor Generalization)** Please provide a detailed case study for the **same task**, showing CODA's full "Thought" and "Action" sequence when commanding (a) the GUI executor vs. (b) the Code agent.
4. **(Re: Case Study)** Can the authors provide a fair version of Figure 6, showing Qwen2.5-VL, UI-TARS-1.5, and CODA on the **same set of tasks**?
5. **(Re: Static Executor)** How do the authors view the risk of the static executor becoming a bottleneck on OOD interfaces? Does the CODA framework have any mechanism to handle this?

---

> ### Author Response · Authors · 2025-11-17
>
> We thank the reviewer for the exceptionally thorough and insightful review. We try to address your concerns below.
> ***
> ### W1-Q1. Critical Missing Baselines
> We have conducted these experiments:
> | Average@8 | Algebra | Biochem | GIS | Astron |
> | --- | --- | --- | --- | --- |
> | Gemini-2.5-Pro + UI-TARS-1.5 | 16.53 | 29.31 | 11.76 | 8.33 |
> | GPT-4o + UI-TARS-1.5 | 18.14 | 30.17 | 9.56 | 7.58 |
> | CODA+UI-TARS-1.5 | 20.16 | 32.23 | 14.17 | 17.05 |
>
> | Pass@8 | Algebra | Biochem | GIS | Astron |
> | --- | --- | --- | --- | --- |
> | Gemini-2.5-Pro+UI-TARS-1.5 | 41.94 | 48.28 | 23.53 | 18.18 |
> | GPT-4o+UI-TARS-1.5 | 38.71 | 51.72 | 17.65 | 15.15 |
> | CODA+UI-TARS-1.5 | 48.39 | 51.72 | 29.41 | 30.30 |
>
> This substantiates our claim and proves that our trainable planner (adapted via decoupled RL) yields superior domain-specific planning compared to proprietary models with limited domain knowledge on OOD softwares. We will merge it into table.1 in the revision.
>
> ***
>
> ### W2-Q2 Automated Judge System
>
> 1. **Rationale for Ensemble Design.** The "distilled model + pre-trained model" ensemble is not necessary to improve the performance of the planner. This is shown in Tab. 3 and Fig. 5, where the 72B-GUI-Judge alone can lead to improvement. Similar to multiple rounds of voting with a single model at high temperature and top-p, the ensemble approach is an engineering design to help improve precision with reduced recall, as requiring two different models to both judge as correct imposes a more stringent standard. We found that higher precision is more important in our RL experiments. We also show in `2` that the teacher model exhibits lower precision but higher recall.
> 2. **Teacher Model Performance.** We will merge this table (Precision/Recall) into Tab 2 in the revision.
> | Method | AgentRewardBench | OSWorld | ScienceBoard |
> | --- | :---: | :---: | :---: |
> | 72B-voting@4 Ensemble | 81.2/76.8 | 79.1/73.0 | 69.5/74.2 |
> | 72B-GUI-Judge | 73.5/79.0 | 76.2/79.4 | 63.7/80.1 |
> | Gemini-2.5-Pro | 79.5/83.6 | 78.4/80.5 | 70.3/81.2 |
> | GPT-4o | 77.4/84.5 | 77.9/81.3 | 69.2/79.9 |
> 3. **Impracticality of Rule-Based Oracles for training.** Most open-ended GUI tasks lack accessible rule-based oracles. For instance, existing benchmarks like OSWorld (300+ tasks) and ScienceBoard (100+ tasks) required immense human effort to construct rule-based verifiers for just a limited number of tasks (which is why they serve solely as evaluation sets). Unlike games where scores provide natural rule-based rewards, judging whether an agent has completed a complex GUI task is inherently challenging without human-annotated verifiers. Therefore, relying on a powerful, general-purpose judge system to predict success is a rational and effective strategy to scale agent training. This approach addresses the challenge of data scarcity and has also been validated in recent works [1, 2].
> ***
> ### W3-Q3 Cross-Executor Generalization
> We provide Fig.9-11 as case study in our revised submission with detailed analysis in Fig's caption, Here are the main points:
>
> Fig. 9 demonstrates how RL refines the planner to generate more informative grounding instructions during training.
>
> Fig. 10 illustrates generalization to code agents. Driven by shared structural patterns with grounding tools, the planner generates more informative instructions and demonstrates strategic invocation—leveraging the external tool specifically when internal command generation fails. This confirms a general enhancement in reliable, structured tool usage.
>
> Fig. 11 highlights efficiency, showing the code agent completing a task in a single line compared to 6 steps of GUI clicks.
> ***
> ### W4-Q4 Case Study on Same Set of Tasks
> In Fig.7, 8, we show all three approaches performing the same task. The results illustrate that UI-TARS-1.5 lacks domain knowledge, while Qwen2.5-VL-72B struggles with grounding.
> ***
> ### W5-Q5 Static Executor Becoming a Bottleneck
> 1. Grounding Executor: Visual grounding is analogous to low-level motor skills like human walk. The primary bottleneck lies not in the grounding capacity, but in how the planner instructs the executor. For example, an executor might fail to process "add current page to collection" on an OOD interface due to layout changes, but it can still precisely execute a visual instruction "click the second star-shaped icon in the upper-left corner." In this perspective, the grounding executor does not become the bottleneck; rather, the planner must adapt to the OOD interface to generate sufficiently descriptive, visually-grounded instructions.
>
> 2. Code Agent Executor: The code agent can be a bottleneck. Facing OOD scenarios (e.g., new languages) inherently requires learning new syntax and commands, representing an inevitable learning requirement distinct from visual grounding and planning.
> ***
> ### References
> [1] UI-TARS-2 Technical Report: Advancing GUI Agent with Multi-Turn Reinforcement Learning.
>
> [2] An illusion of progress? assessing the current state of web agents.

---

> ### Author Response · Authors · 2025-11-28
>
> Dear Reviewer JvNe,
>
> We have included the additional experiments you requested and provided detailed responses to your questions. We hope these updates effectively address your concerns.
>
> Please do not hesitate to let us know if you have any further questions or require additional clarification—we are happy to discuss them at any time.
>
> Thank you for your time and thoughtful review.
>
> Best regards, The Authors

---

### Official Review · Reviewer_gKdc · 2025-10-31

**Soundness:** 4
**Presentation:** 4
**Contribution:** 3
**Rating:** 6
**Confidence:** 3

**Summary:**

This paper presents a decoupled reinforcement learning framework that trains a high-level planner and pair it with a fixed low-level executor for agentic GUI-based computer use tasks. It introduce a two-stage training pipeline for the high level planner to first train software-specialized planner with RL and then train a generalizable planner with SFT on the specialized planner trajectories. Moreover, it alleviates the need for costly human label for RL by proposing an automated judging system powered by vision LLM to provide high quality reward signal automatically. Results on ScienceBoard and OSWorld demonstrate the effectiveness of the planner & executor for both in-domain tasks and out-of-domain generalizability.

**Strengths:**

1. While the formalism of decoupled planner-executor design for agentic framework is well established, this work still provides good insights into further fine-tuning the planner with frozen executor can yield significant benefits in domains like GUI agents, where the low-level grounding can be performed with high accuracy but high-level planning is still challenging with the lack of domain knowledge.
2. The proposed automated judge/reward system largely alleviate the need for human labels, further increase the efficiency of the proposed framework.
3. The experiments and analysis are pretty comprehensive.

**Weaknesses:**

1. While the author states that in stage 1 training they train four specialist models for each software in ScienceBoard. However, as far as I understand, ScienceBoard contains tasks across 6 domains with one software per-domain, so how can four specialized agents cover six softwares? Moreover, as mentioned in ScienceBoard, there are cross-application scenarios which requires more than one software to accomplish the tasks, how are these software specialized models handle cross-application tasks?

**Questions:**

Please see the weaknesses above.

---

> ### Author Response · Authors · 2025-11-17
>
> We sincerely thank the reviewer for the excellent assessment and for accurately summarizing our contribution. We appreciate your insight that "fine-tuning the planner with a frozen executor" is a key solution for the domain knowledge gap in GUI agents. We address your questions regarding the ScienceBoard domains below.
>
> ***
>
> ### W1. Domain Coverage (4 vs. 6) and Cross-Application Scenarios.
>
> We apologize for the ambiguity regarding the domain selection. In our initial submission, we reported results on 4 domains (KAlgebra, ChimeraX, GrassGIS, and Celestia). We excluded Lean and TeXstudio for specific reasons. Below are two example test cases from these domains:
>
> **Lean**
>
> 1. theorem MT_4 [MeasureSpace Ω] [IsProbabilityMeasure (ℙ : Measure Ω)] (X : Ω → ℝ) {p : ℕ} {X : Ω → ℝ} (h₁ : p > 0) (h₂ : Integrable X) (h₃ : ∃ M : ℝ, 𝔼[fun ω => |X ω| ^ p] = M) : Tendsto (fun (x : ℝ) => (x ^ p) * (ℙ {ω : Ω | |X ω| > x}).toReal) atTop (𝓝 0) := by sorry
> 2. theorem ST_1 (X : ZFSet) : ((IsTransitive X) ↔ (X ⊆ powerset X)) ∧ ((IsTransitive X) ↔ ((⋃₀ X : ZFSet) ⊆ X)) := by sorry
>
> **TexStudio**
>
> 1. Find the name of residue id /A:6 in ChimeraX and fill it in the table of the article shown in TeXstudio replacing the placeholder of '???'.
> 2. The f(24) of the Fibonacci sequence listed in TeXstudio is erroneous. Please find the correct answer in KAlgebra and replace the wrong one with it.
>
> The tasks in Lean mainly focus on Mathematical Formal Language, requiring strong mathematical and linguistic reasoning capabilities. This lies outside the scope of our specific focus on decoupled grounding and planning for multi-modal RL. Therefore, we excluded it from our evaluation (notably, Gemini-2.5-Pro and GPT-4o also achieve a 0% success rate when relying solely on screenshot inputs in the Lean domain)
>
> The cross-application scenarios are **exclusively found** in the TeXstudio domain, as many tasks involve retrieving results from other software and entering them into TeXstudio. We sirectly evaluated the pre-trained CODA model (the ckpt we use to report numbers in Table 1 of our work) against other baselines on TeXstudio. As shown in the table below, CODA demonstrates superior performance, proving the cross-application capability of our framework. These results will be merged with Table 1 in the revision.
>
> | Models | TexStudio SR(%) |
> | --- | :---: |
> | Gemini-2.5-Pro | 18.75 |
> | Qwen2.5VL-72B | 12.50 |
> | UI-TARS-1.5 | 0.00 |
> | CODA average@8 | 19.53 |
> | CODA pass@8 | 25.00 |
>
> Crucially, we emphasize that the effectiveness of our method stems from enhancing the planner's fundamental planning and self-reflection capabilities, in addition to acquiring specific domain knowledge. While domain knowledge is software-dependent, the learned reasoning and error-correction logic is highly transferable. This generalizable capability allows the agent to:
> * Handle Cross-Application Logic: Successfully coordinate multi-step workflows in TeXstudio, such as retrieving data from one software and correctly filling it into another.
> * Generalize to OOD Settings: Achieve strong performance on OSWorld (as explicitly evidenced in Table 3 of our submission), proving that the planning strategies learned via decoupled RL remain effective even on novel applications.
> ***

---

> ### Author Response · Authors · 2025-11-25
>
> Dear Reviewer gKdc,
>
> We have included the additional experiments you requested and provided detailed responses to your questions. We hope these updates effectively address your concerns.
>
> Please do not hesitate to let us know if you have any further questions or require additional clarification—we are happy to discuss them at any time.
>
> Thank you for your time and thoughtful review.
>
> Best regards,
>
> The Authors

---

### Official Review · Reviewer_n3de · 2025-10-31

**Soundness:** 3
**Presentation:** 3
**Contribution:** 2
**Rating:** 6
**Confidence:** 4

**Summary:**

This paper introduces CODA, a trainable compositional framework for GUI agents that addresses the challenge of automating complex software tasks requiring both high-level planning and precise execution. Inspired by the functional division between the cerebrum and cerebellum in the human brain, the authors propose decoupling these responsibilities into a learnable Planner (based on Qwen2.5-VL) and a fixed Executor (UI-TARS-1.5 or GUI-Actor). The result is a two-stage training pipeline using decoupled reinforcement learning: (1) a Specialization stage where individual expert planners are trained for specific software using Group Relative Policy Optimization (GRPO) with rewards from an automated judging system, and (2) a Generalization stage where trajectories from all specialists are aggregated to train a unified generalist planner via supervised fine-tuning.

**Strengths:**

The paper is generally well-presented and easy to follow.  The analogy to the human brain's cerebrum-cerebellum division provides an intuitive conceptual framework that helps readers understand the motivation for decoupling planning from execution. The experimental evaluation demonstrates that the presented framework performs strongly better than closed larger models. The fact that the model trained on ScienceBoard shows meaningful performance on the unseen OSWorld benchmark is a good sign as well. The paper also includes comprehensive ablations of all the components and overall I was satisfied with the analysis.

**Weaknesses:**

While CODA works better than closed models would have been good to include other agentic frameworks for comparison. Also Table 1 needs to be explained better. it is not very clear what Average @1, 8... stand for

Minor thing, LVLM acronym is introduced before explaining what is it (first parag of Sec 2)

**Questions:**

See above

---

> ### Author Response · Authors · 2025-11-17
>
> We sincerely thank the reviewer for the positive assessment and for recognizing the intuition behind our cerebrum-cerebellum design. We are glad that the motivation for decoupling planning from execution resonated with you, and we appreciate your constructive suggestions to strengthen the paper. We try to address your concerns below
>
> ***
>
> ### W1-1. Comparison with Other Agentic Frameworks.
>
> We appreciate the suggestion to include broader comparisons. We have evaluated our CODA framework (Qwen2.5-VL-32B Planner + UI-TARS-1.5-7B Executor) against the recent state-of-the-art agentic frameworks Agent-S2 [1] and Agent-S3 [2] on the ScienceBoard benchmark.
>
> | Pass@8 | Algebra | Biochem | GIS | Astron |
> | --- | --- | --- | --- | --- |
> | Agent-S2 [1] | 41.94 | 48.28 | 20.59 | 24.24 |
> | Agent-S3 [2] | 45.16 | 55.17 | 23.53 | 27.27 |
> | CODA+UI-TARS-1.5 | 48.39 | 51.72 | 29.41 | 30.30 |
>
> As shown in the table above, CODA outperforms these strong agentic baselines on most tasks.
>
> - **Unique Value on OOD Software:** While frameworks like Agent-S rely on the inherent knowledge of very large proprietary models, they may struggle with niche, OOD software where domain knowledge is scarce.
> - **Benefit of Decoupled RL:** This comparison highlights the core contribution of our work: the ability to perform Decoupled Reinforcement Learning at zero human cost. By enabling the planner to actively interact with and learn from the specific dynamics of novel software (guided by a general judge), CODA allows for effective adaptation on OOD software where human data is unavailable, surpassing generalist agentic frameworks.
>
> ***
>
> ### W1-2 Clarification on Table 1 Metrics (Average/Pass @1, @8)
>
> We apologize for the ambiguity in the initial draft.
>
> - **Reason for @8:** As noted in [2], GUI agents often exhibit high variance due to multi-turn interactions and dynamic virtual environments. To report more reliable and robust performance, we conduct 8 independent inference runs for each task.
>     - **Pass@8:** The probability that at least one of the 8 runs is successful.
>     - **Average@8:** The average success rate across the 8 runs.
> - **Reason for @1:** To ensure a fair comparison with the official ScienceBoard leaderboard (which reports single-run results), we cite their official numbers and label them as Average@1.
> We will update the caption of Table 1 to explicitly define these metrics in the revision.
>
> ***
>
> ### W2 Acronym before explaination
>
> Thank you for pointing this out. We explicitly define LVLM (Large Vision-Language Model) upon its first appearance in Section 2 in the revision.
>
> ***
>
> ### References
>
> [1]. Agent S2: A Compositional Generalist-Specialist Framework for Computer Use Agents.
>
> [2]. Agent S3: Approaching Human-level Computer Use with Wide Scaling.

---

> > ### Author Response · Authors · 2025-11-26
> >
> > Dear Reviewer n3de,
> >
> > We have included the additional experiments you requested and provided detailed responses to your questions. We hope these updates effectively address your concerns.
> >
> > Please do not hesitate to let us know if you have any further questions or require additional clarification—we are happy to discuss them at any time.
> >
> > Thank you for your time and thoughtful review.
> >
> > Best regards, The Authors

---

### Official Review · Reviewer_XXbi · 2025-11-08

**Soundness:** 2
**Presentation:** 2
**Contribution:** 2
**Rating:** 4
**Confidence:** 3

**Summary:**

CODA introduces a two-component system—Cerebrum (planner) and Cerebellum (executor)—to overcome the common trade-offs between high-level planning and low-level execution in autonomous GUI agents. The paper claims to outperform prior systems by decoupling these tasks, allowing the planner to adapt via reinforcement learning, while the executor remains fixed. It presents a two-stage training process that first specializes the planner for specific software and later generalizes it across domains. Despite these grand claims, the paper largely fails to address key concerns, both in terms of practicality and theoretical soundness.

**Strengths:**

Generalization Across Software: By leveraging a specialist-to-generalist approach, CODA achieves strong generalization across novel software environments. Its ability to adapt to different software systems without requiring human-labeled data is a significant improvement over many existing systems.

**Weaknesses:**

Ambitious Design: The idea of decoupling high-level planning from low-level execution is an interesting attempt to mimic human cognition. The use of a "Cerebrum" and "Cerebellum" model, while conceptually engaging, feels overly complex for the problem at hand. It’s as though the authors were trying too hard to sound cerebral, when a simpler solution might suffice.

Potential Overfitting: While the system is trained to generalize, there is a risk of overfitting during the specialization phase, especially when only a limited set of software environments are used. Ensuring that the generalist model is truly robust and does not perform poorly on unseen tasks remains a challenge.

Questionable Innovation: The “Cerebrum-Cerebellum” split is framed as a revolutionary idea. Yet, many systems already rely on modular planning-execution pipelines. The authors' framework adds little to the discussion in terms of novel methodologies. It feels like the paper is rebranding an old idea under a shiny new name.

**Questions:**

The system is based on reinforcement learning—what happens when you encounter tasks that are outside the scope of the training set? Is there an inherent limitation in CODA’s ability to generalize, or can it truly handle novel scenarios?

---

> ### Author Response · Authors · 2025-11-17
>
> Thank you for your review and constructive feedback, we try to address your concerns below.
>
> ***
>
> ### W1-1. The decoupled approach overly complex for the problem at hand.
>
> We respectfully clarify that the decoupled design is not merely for complexity, but a necessity driven by the difficulty of the GUI navigation task.
>
> - **Empirical Evidence:** As shown in the OSWorld and ScienceWorld leaderboards, single end-to-end models lag behind modular, agentic approaches [1-3]. Simpler solutions have proven insufficient for handling the long-horizon reasoning and precise grounding required in computer use.
> - **Design Rationale:** By decoupling the planner (Cerebrum) from the executor (Cerebellum), we allow the planner to focus on high-level reasoning while the executor handles low-level grounding. This separation is crucial for our reinforcement learning, enabling us to update the planner's policy efficiently without destabilizing the grounding capabilities.
>
> ***
>
> ### W1-3. Novelty of the "Cerebrum-Cerebellum" Split (vs. Existing Pipelines)
>
> This is a critical misunderstanding we wish to clarify. While modular *inference* pipelines exist, **CODA represents a fundamental shift from "Inference-only" to "Trainable" agentic framework.**
>
> - **Existing Systems:** As you noted, many systems [1-3] use modular pipelines. However, these are predominantly proprietary, **inference-only** frameworks (wrapping closed-source models like GPT-4o) that cannot update their parameters based on interaction with desired software.
> - **Our Innovation:** To our knowledge, CODA is the first framework to enable **decoupled Reinforcement Learning** for a modular planning-execution pipeline. This allows the agent to *learn* from interaction and improve over time. Training a multi-agent system is non-trivial. The primary bottleneck in novel software is not the low-level grounding (e.g., "how to click an icon"), but the lack of high-level domain knowledge. By freezing the robust executor (Cerebellum) and focusing RL solely on the planner (Cerebrum), we force the training process directly target this reasoning gap. This design allows the planner to efficiently learn novel workflows (acquiring domain knowledge). Our experiments confirm that this strategy is more effective than training a monolithic, end-to-end agent.
> - **Results:** This trainable nature is why CODA outperforms larger, proprietary models (e.g., GPT-4o, Gemini-2.5-Pro) using zero human-annotated data. The innovation lies in the training methodology (Decoupled RL with a reliable Judge) rather than just the modular architecture itself.
>
> ***
>
> ### Q1&W1-2. Potential overfitting and generalization of CODA.
> To address the concern regarding tasks outside the training scope, our empirical results already demonstrate that CODA's **Decoupled RL approach** truly handles novel scenarios effectively, overcoming the overfitting limitations inherent in standard Behavior Cloning. As shown in table below (coped from Tab.3) and Fig.4b, we conduct training specifically on the **ChimeraX software** (BioChem domain). When evaluated on the **completely unseen OSWorld benchmark** (comprising software not encountered during training), the SFT (Behavior Cloning) performance drops to 10.25% due to overfitting. In contrast, our RL-trained model achieves significant gains on these unseen tasks (improving from 12.47% to 19.39%). This confirms that our decoupled RL framework enables the planner to learn **transferable planning logic** effective on novel software, rather than merely memorizing specific trajectories from the training set.
>
> | Model | BioChem (In-Domain) | OSWorld (OOD) |
> | :--- | :---: | :---: |
> | **Baseline (Qwen2.5VL+GUI-Actor)** | 17.24 | 12.47 |
> | **RL Training (72B-Judge-Ensemble)** | **30.17** | **19.39** |
> | **SFT (Behavior Cloning)** | 28.02 | 10.25 |
>
> Second, regarding our design philosophy: Our work does not aim to build a model that can directly generalize to OOD software without training.  Instead, our work aims to propose a framework that performs reinforcement learning via interaction with software, featuring a modular design and a decoupled training approach with zero human effort. Combined with a reliable automatic judge system that provides reward signals, our framework can achieve strong performance on OOD software. We emphasize again that our framework requires no human annotation, so there is no “training set” (if “training set” implies a human-labeled dataset). For any novel software without human annotation, our framework can train the model via interaction to achieve improvement. We selected ScienceBoard as our evaluation benchmark specifically to evaluate our framework on truly novel software without available human annotation.
> ***
>
> ### References
>
> [1] Agent S: An Open Agentic Framework that Uses Computers Like a Human.
>
> [2] Agent S2: A Compositional Generalist-Specialist Framework for Computer Use Agents.
>
> [3] CoAct-1: Computer-using Agents with Coding as Actions.

---

> > ### Author Response · Authors · 2025-11-25
> >
> > Dear Reviewer XXbi,
> >
> > We have included the additional experiments you requested and provided detailed responses to your questions. We hope these updates effectively address your concerns.
> >
> > Please do not hesitate to let us know if you have any further questions or require additional clarification—we are happy to discuss them at any time.
> >
> > Thank you for your time and thoughtful review.
> >
> > Best regards,
> > The Authors

---

### Author Response · Authors · 2025-12-04
**General Response and Summary of Rebuttal Updates**

Dear Area Chair(s),

We sincerely appreciate the time and effort reviewers have invested in evaluating our work. Here, we summarize the key facts regarding our submission and the constructive author-reviewer discussion, which we hope will assist you in making recommendations.

During the discussion phase, we actively participated in response with all reviewers. We have completed **all requested experiments**—most notably adding the critical proprietary baselines and validating our decoupled RL framework—which provide strong empirical support for our claims. We summarize the discussion highlights in the table below.

| Reviewer | Initial Rating | Concerns / Request | Responses / Extra Exps |
| :--- | :--- | :--- | :--- |
| **JvNe** | 4 | Missing "Gold Standard" baseline (GPT-4o/Gemini + UI-TARS); questions on Judge System precision; doubts on cross-executor generalization. | **Added GPT-4o & Gemini-2.5-Pro + UI-TARS baselines (CODA outperforms)**; provided Judge precision/recall metrics; added Fig. 9-11 case studies showing generalization to Code Agent. |
| **XXbi** | 4 | "Cerebrum-Cerebellum" design complexity; concerns about overfitting to training domains; novelty of "trainable" pipeline. | **Clarified that modular design is essential** for complex tasks (standard in SOTA); distinguished CODA from existing **inference-only** approaches by introducing **decoupled RL** via interaction with software as first trainable agentic framework; validated efficacy on unseen tasks. |
| **n3de** | 6 | Comparison with other agentic frameworks (Agent-S); clarification on Table 1 metrics (@1 vs @8). | **Compared against Agent-S2 and Agent-S3 (CODA outperforms)**; clarified Average@8 vs Pass@8 definitions; addressed acronym clarifications. |
| **gKdc** | 6 | Domain coverage in ScienceBoard; handling of cross-application scenarios (e.g., TeXstudio). | **Provided TeXstudio results** demonstrating strong cross-app performance; clarified domain selection rationale (noting Lean is the **sole** exclusion as it is a language for **formal mathematical proofs**, which even Gemini-2.5-Pro cannot handle and beyond the research domain of GUI-Agent). |

**Discussion Outcomes and Progress:**

1.  **Critical Baselines & SOTA Validation (Addressing JvNe & n3de):**
    We addressed the primary concern regarding the lack of a "Gold Standard." New experiments confirm that **CODA (Open-Source Planner) outperforms proprietary planners (GPT-4o, Gemini-2.5-Pro)** when paired with the same executor. We also demonstrated superior performance over state-of-the-art agentic frameworks **Agent-S2 and Agent-S3**.

2.  **Generalization & Cross-Application Capabilities (Addressing XXbi & gKdc):**
    To address concerns about overfitting, we evaluated the model on **TeXstudio**, a domain requiring cross-application interaction. CODA significantly outperforms baselines in these scenarios, validating that the planner learns **generalized reasoning logic** rather than merely memorizing domain-specific trajectories.

3.  **Clarification on Technical Design (Addressing JvNe & XXbi):**
    We provided the requested **Precision/Recall metrics** for the teacher models, justifying our ensemble design for stable RL signals. Furthermore, new case studies (**Fig. 9-11**) illustrate how the planner adapts its high-level "Thought" process to control an unseen **Code Agent**, confirming the effectiveness of the decoupled architecture.

We hope this summary provides a clear picture of the substantial improvements made during the rebuttal phase. We remain fully committed to incorporating these valuable insights into our final revision to further elevate the quality of our work.

Best regards,
Authors of Submission #3273

---

### Meta-Review · Area_Chair_EH4S · 2026-01-07

**Summary:**

This paper proposes CODA, a trainable planner–executor framework for GUI agents with a two-stage specialist-to-generalist training pipeline based on decoupled reinforcement learning.

**Reviewer Concerns:**

Reviewer opinions were mixed, with scores clustered around the acceptance threshold. Concerns centered on missing critical baselines, the novelty of the approach relative to prior modular or hierarchical agents, the reliability of the automated judge system, and the strength of generalization claims—particularly cross-executor and cross-application transfer.

During the rebuttal and discussion phase, the authors made an effort to address some major concerns. They added the previously missing “gold-standard” proprietary baselines (GPT-4o and Gemini-2.5-Pro paired with the same executor), and comparisons to open-source agentic frameworks (Agent-S2/S3). Concerns about overfitting and limited domain coverage were mitigated by new experiments on cross-application scenarios (e.g., TeXstudio), which demonstrate that CODA generalizes beyond single-software settings.

But questions about the conceptual novelty relative to prior planner–executor or hierarchical RL frameworks, and about the reliability of the automated judge system, remain to some degree.

**Reviewer Scores:**

I believe the reviewers will maintain their scores.

---

### Decision · Program_Chairs · 2026-01-26

Reject